

# A general database of hydrometeor single scattering properties at microwave and sub-millimetre wavelengths

Patrick Eriksson[1], Robin Ekelund[1], Jana Mendrok[1], Manfred Brath[2], Oliver Lemke[2], and Stefan A. Buehler[2]

[1]Department of Space, Earth and Environment, Chalmers University of Technology, Gothenburg, Sweden
[2]Meteorological Institute, Department of Earth Sciences, Center for Earth System Research and Sustainability, Universität Hamburg, Germany

*Correspondence to:* Patrick Eriksson (patrick.eriksson@chalmers.se)

**Abstract.** A main limitation today in simulations and inversions of microwave observations of ice hydrometeors (cloud ice, snow, hail ...) is the lack of data describing the interaction between electromagnetic waves and the particles. To improve the situation, the development of a comprehensive dataset of such "scattering properties" has been started. The database aims at giving a broad coverage in both frequency (1 to 886 GHz) and temperature (190 to 270 K), to support both passive and active

current and planned measurements, and to provide data corresponding to the full Stokes vector. This first version of the database is restricted to totally random particle orientation. Data for 34 particle sets, i.e. habits, have been generated. About 17 of the habits can be classified as single crystals, three habits can be seen as heavily rimed particles, and remaining habits are aggregates of different types, representing e.g. snow and hail. The particle sizes considered vary between the habits, but a maximum diameter of 10 and 20 mm are typical values for the largest single crystal and aggregate particles, respectively, and the number of

sizes per habit is at least 30. Particles containing liquid water are also inside the scope of the database, but this phase of water is so far only represented by a liquid sphere habit. The database is built upon the netCDF4 file format. Interfaces to browse, extract and convert data for selected radiative transfer models are provided in Matlab and Python. The database and associated tools are publically available from Zenedo (https://doi.org/10.5281/zenodo.1175572 and https://doi.org/10.5281/zenodo.1175588, respectively). Planned extensions include non-spherical rain drops, melting particles and a second orientation case that can be

denoted as azimuthally random.

## 1 Introduction

### 1.1 Motivation

Atmospheric particles absorb, emit and scatter electromagnetic radiation. The optical properties governing these processes are not easily determined for an arbitrary shaped particle (e.g. Min et al., 2006), and radiative transfer simulations must frequently

make use of tabulated values to limit the calculation burden. At microwave wavelengths, the targets of a database of particle optical properties are hydrometeors, i.e. cloud and precipitating particles that consist of condensed water. The need to consider particle shape in detail increases with frequency and is the highest for hydrometeors containing ice, so this is the case



where tabulated data are most urgently required. Aerosols can in general be neglected for microwave radiative transfer, but measurements of volcanic ash are another potential application of sub-millimetre radiometry (Baran, 2012a).

Microwave particle optical properties, also often denoted as scattering data, are most obviously needed for measurements of precipitation and cloud properties. Active instruments, i.e. radars, perform such measurements by detecting back-scattering of hydrometeors. Radars targeting rain operate at relatively low frequencies and are well established (Doviak and Zrnic, 2014). For example, ground-based networks of precipitation radars are today found in many countries. The sensitivity to cloud particles and snow increases with frequency. The highest standard radar frequency today is 94 GHz (Kollias et al., 2007), used both from ground (e.g Hogan et al., 2000), aircraft (e.g. Bouniol et al., 2010) and by the first space-based "cloud radar" onboard CloudSat launched 2006 (Stephens et al., 2008) as well as the CPR instrument onboards of ESA's EarthCare mission (Illingworth et al., 2015).

The situation is more complex for passive instruments. Extinction due to scattering is the main process by which rain and ice particles are sensed, but in general all the optical properties must be considered to understand the radiance signal of hydrometeors (e.g. Liou, 2002). The relative importance of absorption and emission is the highest for particles that are small compared to the wavelength, and also higher for liquid particles, compared to ice ones, due to differences in refractive index. The present set of operational microwave sensors cover frequencies up to about 190 GHz (e.g. Weng et al., 2012). The main hydrometeor products of these measurements are today cloud liquid water and rain (Boukabara et al., 2011). Ice hydrometeor retrievals exist (e.g. Boukabara et al., 2013), but are less established and are often purely empirical using satellite radar data for "training" (Holl et al., 2014; Piyush et al., 2017).

At present, the sub-millimetre part of the microwave region is primarily used for limb sounding. These instruments target mainly the strato- and mesosphere, but can provide data on e.g. humidity and cloud ice down to about 10 km (Wu et al., 2008; Eriksson et al., 2014). Sub-millimetre measurements dedicated to the troposphere are so far only at hand by some aircraft instruments (Evans et al., 2012; Fox et al., 2017; Brath et al., 2018). Operational sensors, i.e. such used for weather forecasting, making use of sub-millimetre wavelengths will be at hand from about 2022 (Kangas et al., 2012). The addition of this wavelength region will improve the general sensitivity of passive microwave measurements, particularly regarding ice hydrometeors. Adding this wavelength region should provide a better constraint on particle size of frozen hydrometeors and, not only the sensitivity, but also the accuracy of ice hydrometeor mass retrievals should be improved (Evans and Stephens, 1995; Buehler et al., 2007a).

Scattering data are not only important for the direct measurement of hydrometeors, but also essential to characterise the interference of clouds and precipitation on measurements targeting other quantities. One such example is the "cloud screening" applied in humidity retrievals based on channels around the 183 GHz water vapour transition (Buehler et al., 2007b). The addition of sub-millimetre channels should facilitate this cloud screening. Passive microwave data, particular using channels on the low frequency end, are also used to estimate ocean surface variables, and even these retrievals are affected by hydrometeors. The characterisation of this interference can even involve to understand polarisation effects (Adams et al., 2008).

In summary, microwave sensors offer good information on cloud and precipitation, on the same time as surface variables, humidity and some other gases can be estimated with a relatively low cloud interference (Kunzi et al., 2011). These advanta-



geous aspects are today heavily used for both climate studies and weather prediction. However, the full potential of existing data is far from fully exploited (e.g. Guerbette et al., 2016), on the same time as preparations for future measurements are needed (e.g. Buehler et al., 2012; Birman et al., 2017). In both cases, progress relies on that rigorous simulations can be performed, but the quality of microwave simulations is today strongly limited by a lack of hydrometeor scattering data (Geer and

Baordo, 2014). Liquid particles can to a large extent be assumed to be spherical, and the scattering properties can be calculated efficiently by Mie theory. On the other hand, both cloud and snow ice hydrometeors exhibit a huge variability in shape (e.g. Lynch et al., 2002; Garrett et al., 2015; O'Shea et al., 2016), and to derive their scattering data is a demanding calculation task. These complications have resulted in that the coverage of ice hydrometeor microwave scattering data so far has been patchy.

## 1.2 Single scattering data

To be of general character, scattering data must cover a number of dependencies, as discussed in later sections. For a given particle, the scattering properties vary with frequency, polarisation, temperature and orientation. Furthermore, particle size and shape play a significant role, where the comprehensive description of particle shape is less clear. Several measures on particle size are being used, but that represents a small complication in comparison to morphology classification. Traditional schemes, such as Magono and Lee (1966), offer a good starting point regarding the classification of pristine crystals. Ice hydrometeors

above about $200\,\mu m$ in maximum dimension tend to be aggregates of crystals (Schmitt and Heymsfield, 2014), hence aggregates should in general dominate the scattering signature in microwave data, which is mainly determined by particles larger than $100\,\mu m$ (Eriksson et al., 2008). However, there is no established manner to categorise this particle class. So far aggregates are described in relatively crude manners, by e.g. effective density and degree of riming. A further complication is that both liquid and ice water can be found in a particle simultaneously. The so called radar bright-band is normally taken as a

manifestation of the presence of melting ice hydrometeors (Sassen et al., 2007).

A review of existing microwave scattering data was made by Eriksson et al. (2015). At that point, the most general collection was provided by Liu (2008) despite it having been limited to totally random particle orientation, frequencies below 340 GHz and low complexity in particle morphology. The only source to data covering sub-millimetre wavelengths was Hong et al. (2009), but it was found to be based on an outdated ice refractive index parameterisation. More complex ice particles,

i.e. particles of aggregate type, had been considered (Tyynelä and Chandrasekar, 2014) but then providing data with limited frequency coverage and mainly targeting radar applications. More recently, Kuo et al. (2016) presented data for a large number of habits, including many aggregate realisations, but the publicly available data lack detailed information and provided only parameters applicable in simplified radiative transfer. The data of Lu et al. (2016) have interesting features regarding particle shapes and orientation, but are not well suited for passive microwave applications since based on an obsolete, and inaccurate,

refractive index parameterisation (see Sec. 4.1 for details). The data of Ding et al. (2017), which are an updated and extended version of the database by Hong et al. (2009), cover 1 to 874 GHz, at multiple temperatures, and can thus be considered to be first to provide a full coverage of the microwave region, but are still limited to totally random orientation and relatively simple habits.





Besides the production of databases, there is a number of investigations of relationships between particle properties and optical properties. For example, the data of Ding et al. (2017) were compared to the scattering properties of more complex, snow-like, particles by Baran et al. (2018). The impact of riming on scattering properties was studied by Leinonen and Szyrmer (2015), using a triple frequency approach. A clear shift in radar signature was observed as riming was gradually applied to
pristine aggregates. Also data for melting particles have been reported (Ori et al., 2014; Johnson et al., 2016), showing an increase in extinction and back-scattering at increasing melting fractions.

In summary, important progress has been made, but there are still important gaps in available data. Based on the findings in Eriksson et al. (2015), a decision was taken to gradually build up a database of microwave scattering properties, with the long term goal of providing a comprehensive source of such data. The ambition is to provide data of general character,
e.g. supporting both passive and active observations involving polarimetric information in a consistent manner. A secondary objective is to ensure that advanced scattering solvers (e.g. RT4 by Evans and Stephens (1995) and MC by Davis et al. (2005)) as included in the ARTS-2 forward model (Eriksson et al., 2011a) can be fully utilised. In the following, we will hence refer to the data compilation as the "ARTS scattering database".

The first version of the database is presented below. The database is not complete, but already this version is more extensive
than earlier datasets. The frequency coverage is high, very similar to the one of Ding et al. (2017), and multiple temperatures are covered. Full polarimetric data are given. A special emphasis has been given to improve on the representation of aggregates; 34 ice hydrometeor habits of totally random orientation are included, with 16 of these being of aggregate type. Pure liquid particles are also covered, in the form of liquid spheres. Data for more specific orientations (such as particles with a preference towards the horizontal plane) are left for the next database version. Non-spherical liquid particles and melting particles are
other planned database extensions.

An overview of the database content is found in Sec. 2. A number of software tools have been used to generate the database, and those are presented in Sec. 3. The various microphysical assumptions are summarised in Sec. 4. Example results are found in Sec. 5. Data format, interfaces and availability are discussed in Secs. 6 and 7. Finally, conclusions are found in Sec. 8.

## 2  Database content

This section gives an overview of the optical properties provided, and the coverage of the database in terms of frequency, size, temperature and habits.

### 2.1  Radiative properties

The optical properties of a particle can be reported in several manners. The ARTS database assumes that electromagnetic
radiation is described using the Stokes formalism and provides the extinction matrix, the absorption vector and the phase





matrix. These quantities are most easily introduced by considering the basic radiative transfer equation:

$$\frac{\mathrm{d}I(\nu,\boldsymbol{r},\boldsymbol{n})}{\mathrm{d}s} = \ -\mathbf{K}(\nu,\boldsymbol{r},\boldsymbol{n})I(\nu,\boldsymbol{r},\boldsymbol{n}) + \mathbf{a}(\nu,\boldsymbol{r},\boldsymbol{n})B(\nu,\boldsymbol{r})$$
$$+ \int_{4\pi}\mathbf{Z}(\nu,\boldsymbol{r},\boldsymbol{n}',\boldsymbol{n})I(\nu,\boldsymbol{r},\boldsymbol{n}')\mathrm{d}\boldsymbol{n}', \tag{1}$$

where $I$ is the Stokes vector of the radiance, $\nu$ is the frequency, $\boldsymbol{r}$ is the position, $\boldsymbol{n}$ is the propagation direction of concern, $s$ is

the distance along $\boldsymbol{n}$, $\mathbf{K}$ is the extinction matrix, $\mathbf{a}$ is the absorption vector, $B$ is the Planck function, $\mathbf{Z}$ is the phase matrix and $\boldsymbol{n}'$ represents the propagation direction of radiation scattered towards $\boldsymbol{n}$. See Mishchenko et al. (2002) for an exact definition of the Stokes vector assumed, as well as for details regarding assumptions and relationships implied by the equation. Note that $\mathbf{a}$ can be derived from $\mathbf{K}$ and $\mathbf{Z}$, but is included in the database anyways in order to avoid numerical issues and ensure good accuracy of all three main quantities.

To be clear, $\mathbf{K}$, $\mathbf{a}$ and $\mathbf{Z}$ in Eq. (1) represent bulk properties while the database provides these quantities on a particle basis, which are dependent on particle size and dielectric properties. Furthermore, the scattering is assumed to be incoherent between particles, and the bulk properties are simply obtained by summing up $\mathbf{K}$, $\mathbf{a}$ and $\mathbf{Z}$ of all particles inside the considered atmospheric volume. Information allowing radiative transfer with the full Stokes vector (length four) is provided. However, transformations are used in order to save storage space, and some processing of the data is needed (Sec. 6.1) before it can be

applied in Eq. (1).

The radiative transfer equation is not always expressed as in Eq. (1), but it should be possible to derive the quantities found in other versions of the equation from the optical properties provided. For example, some "scattering solvers" operate with the single scattering albedo and the asymmetry parameter. These scalar quantities can be calculated from the elements in $\mathbf{K}$, $\mathbf{a}$ and $\mathbf{Z}$ matching the first Stokes component. The main quantity for simulations of radar measurements is back-scattering, which

essentially is the value of $\mathbf{Z}$ representing the backward direction. For further details around these derived properties, see the database technical report (Ekelund et al., 2018).

## 2.2  Particle orientation

The database considers orientation averaged quantities only. In general, using the scattering matrix $\mathbf{Z}$ as example, orientation averaged quantities are calculated as

$$\mathbf{Z}_{\mathrm{o}}\left(\boldsymbol{n}',\boldsymbol{n}\right) = \int\limits_{0}^{2\pi}\int\limits_{0}^{\pi}\int\limits_{0}^{2\pi} p_{\alpha}(\alpha)\,p_{\beta}(\beta)\,p_{\gamma}(\gamma)\,\mathbf{Z}\left(\boldsymbol{n}',\boldsymbol{n},\alpha,\beta,\gamma\right)\sin\beta\,\mathrm{d}\alpha\,\mathrm{d}\beta\,\mathrm{d}\gamma \tag{2}$$

where $\mathbf{Z}_{\mathrm{o}}$ is the orientation averaged scattering matrix, $\alpha$, $\beta$ and $\gamma$ are here the three Euler angles describing the orientation of particles, and $p_j$ are probability density functions describing the distribution of particle orientation.

The database currently only covers particles in one orientation mode: totally random, defined as the distribution where all particle orientations are equally probable. That is, $p_{\alpha}$, $p_{\beta}$ and $p_{\gamma}$ are all uniform distributions. The database is designed and

intended to also include what can be denoted as azimuthally random orientation. In this orientation case, the angles $\alpha$ and $\gamma$ are fully random as for the totally random case, while the data is evaluated at discrete tilt angles $\beta$ (i.e. $p_{\beta}$ is a Dirac delta





function). It turns out that azimuthally random orientation is much more costly both with respect to calculation burden and data storage, and this orientation mode is left for next database version.

Also, a reference particle alignment must be defined, as it affects both our definition of aspect ratio (Sec. 2.3) and data for azimuthally random orientation. The reference alignment should be linked to how ice particles on average are oriented in the atmosphere. Accordingly, we align the particle with its maximum principal moment of inertia axis parallel to the z-axis. Furthermore, the minimum principal axis is aligned along the x-axis. This approximates the mean alignment of falling particles without performing extensive aerodynamics simulations. The definition is also stable in the sense that particular features, such as a narrow arm of the particle, will have less effect than the overall mass distribution of the particle.

## 2.3 Size and shape descriptors

Quantitatively characterising size and shape of irregularly shaped particles, e.g. of aggregates, is a non-trivial task. We provide several descriptive values on particle size. First of all, particle mass $m$ is reported. Secondly, the volume equivalent diameter is given, which is related to mass by

$$D_{\mathrm{veq}} = \left(\frac{6m}{\pi\rho}\right)^{1/3}, \tag{3}$$

where $\rho$ is the actual density of the material found in the particle. Note that $D_{\mathrm{veq}}$ relates to the diameter of volume-equivalent ice sphere, not of a mass equivalent melted sphere. The size parameter applied, $x$, is defined with respect to $D_{\mathrm{veq}}$:

$$x = \pi D_{\mathrm{veq}}/\lambda, \tag{4}$$

where $\lambda$ is the wavelength.

Another common parameter is the maximum dimension $D_{\mathrm{max}}$ (sometimes used interchangeably with the maximum diameter), for which no standard convention exists. For instance, photography-based in-situ measurements often derive the maximum dimension by the largest distance between two pixels in a 2D-image. Such a definition is highly dependant of the particle orientation with respect to the camera. In the ARTS database $D_{\mathrm{max}}$ is defined as the diameter of the minimum subscribing sphere, which takes the whole 3D-structure into account. However, ice particles are often highly irregular, and since $D_{\mathrm{max}}$ is a measure of a particle's extreme points, it is an inherently ambiguous and ill-defined parameter. Hence, it should be considered with caution, and the mass or $D_{\mathrm{veq}}$ should in general be preferred.

Similarly, the aspect ratio (AR) is defined differently by different authors. In general it provides a measure of the non-sphericity of the particle of concern, thus affecting the particle's preferred orientation and its scattering properties (polarimetric quantities in particular). The definition of AR is ambiguous and ill-defined, in similar manners as $D_{\mathrm{max}}$. Our definition of AR is linked to how the particle is oriented with respect to the horizontal plane. Based on the particles' reference orientation outlined in Sec. 2.2, we define AR as the ratio of the maximum particle span found in the XY-plane to the span along the z-axis.

Axial ratio is sometimes used interchangeable with aspect ratio. We restrict the use of this term to describe the shape of hexagonal prism columns and plates, with axial ratio defined as $L/(2a)$ where $L$ is the height of the prism and $2a$ its diameter. With this definition, columns/plates have an axial ratio above/below unity.



The database contains also a simplistic measure of aerodynamic cross-sectional area for each particle. The variable stored is the diameter of a circle with an area equal to the the cross-sectional area when viewing along the zenith angle. For this calculations the particle orientation is assumed fixed in its reference position, i.e. there is a disparity compared to how the scattering properties are calculated, where totally random orientation is assumed.

## 2.4 Habits

The term "habit" is frequently used in the context of particle shape. We define habit as a set of particles with a common basic morphology and following a rule on how the morphology varies with size. As an example, a common habit assumption is hexagonal columns, having an aspect ratio that depends on size according to an analytic expression. Furthermore, the relationship between maximum diameter and mass of a habit is commonly parameterised by two constants, $a$ and $b$, as

$$m = aD_{\max}^{b}. \tag{5}$$

The $a$ and $b$ values reported below assume SI units. As a reference, a habit consisting solely of ice spheres has $a = 480$ and $b = 3$. The ambition is that the database habits shall follow Eq. (5) as far as possible, but an exact fit is normally not possible, e.g. when using externally generated shape data. For $b < 3$, Eq. (5) can only be considered as valid above some threshold size, normally in the order of $100\,\mu\text{m}$, because Eq. (5) otherwise requires unphysically high densities ("superdensity") for $D_{\max}$ below the threshold size.

In total, 34 habits divided into several subgroups are provided (Table 1). Each habit is assigned an unique Id-number. Visualisations of the single crystal and aggregate habits are displayed in Figs. 1 and 2, respectively. The $a$ and $b$ parameters given in Table 1 were calculated by a least squares fit between $log(m)$ and $\log(D_{\max})$, including only particles having a $D_{\max} \geq 200\,\mu\text{m}$. The limitation in size was applied to not pass the threshold size discussed in the paragraph above.

As mentioned, no totally general classification of particle shape exists, but for user friendliness the database is structured according to a rough categorisation of the habit type. The top category is "phase": liquid, ice or melting. The ice and melting habits are further divided between single crystals and aggregates. This classification level is not applied for liquid particles. Both single crystals and aggregates are classified as either pristine or rimed. Table 2 gives an overview of this categorisation. Further information on how the particles were generated is provided in Sec. 3.1.

## 2.5 Size grid

The number and range of sizes are not identical between the habits due to a number of reasons. For some habits, mainly aggregates, it is not possible to generate shape data on a pre-defined size grid. Certain habits have been provided by third-party sources, where control of the size grid is limited. Furthermore, the methodologies of designing the habits have evolved over the time of the database creation. Lastly, it makes sense to target different size ranges with single crystal and aggregate habits. However, the general strategies to set the size grids are similar.

The minimum number of sizes was set to 30, but single crystal data are still provided for 35 to 45 sizes. Aggregates have mainly 35 sizes. Maximum size ranges were defined with respect to both $D_{\max}$ and $D_{\text{veq}}$ (Table 3). The range limits were set



**Table 1.** Habits included in the first version of the database. Habits marked with $^*$ are calculated using Mie theory. The last column displays the software or source used to created the shape data of the given habit, with abbreviations being SFTK (SnowFlake ToolKit, Sec. 3.1.1), RC (RimeCraft, Sec. 3.1.2), RSP (Recreated Shape Data, Sec. 4.2) and ESP (External Shape Data, Sec. 4.3.1). See the text for further details.

| Habits | Id | $D_{\max}$ [$\mu$m] | $D_{\mathrm{veq}}$ [$\mu$m] | No. of sizes | $a$ | $b$ | Software used |
|---|---|---|---|---|---|---|---|
| **Ice:** | | | | | | | |
| **Single crystals:** | | | | | | | |
| **Pristine:** | | | | | | | |
| Plate type 1 | 9 | 13 – 10,000 | 10 – 2,596 | 45 | 0.76 | 2.48 | RSP |
| Column type 1 | 7 | 14 – 10,000 | 10 – 1,815 | 45 | 0.037 | 2.05 | RSP |
| Thin plate | 16 | 25 – 5,059 | 10 – 2,000 | 35 | 30 | 3.00 | RSP |
| Thick plate | 15 | 16 – 3,246 | 10 – 2,000 | 35 | 110 | 3.00 | RSP |
| Block column | 12 | 13 – 2,632 | 10 – 2,000 | 35 | 210 | 3.00 | RSP |
| Short column | 13 | 17 – 3,303 | 10 – 2,000 | 34 | 110 | 3.00 | RSP |
| Long column | 14 | 24 – 4,835 | 10 – 2,000 | 35 | 34 | 3.00 | RSP |
| Sector snowflake | 3 | 20 – 12,000 | 20 – 1,415 | 34 | 0.00081 | 1.44 | RSP |
| Ice sphere$^*$ | 24 | 1 – 50,000 | 1 – 50,000 | 200 | 480 | 3.00 | Mie |
| ICON cloud ice | 27 | 13 – 10,000 | 10 – 2,929 | 45 | 1.6 | 2.56 | SFTK |
| GEM cloud ice | 31 | 10 – 3,088 | 10 – 3,000 | 45 | 440 | 3.00 | SFTK |
| 6-bullet rosette | 6 | 16 – 10,000 | 10 – 2,371 | 45 | 0.48 | 2.42 | RSP |
| 5-bullet rosette | 2 | 17 – 10,000 | 10 – 2,231 | 45 | 0.4 | 2.43 | SFTK |
| Perpendicular 4-bullet rosette | 10 | 18 – 10,000 | 10 – 2,071 | 45 | 0.32 | 2.43 | SFTK |
| Flat 4-bullet rosette | 11 | 18 – 10,000 | 10 – 2,071 | 45 | 0.32 | 2.43 | SFTK |
| Perpendicular 3-bullet rosette | 4 | 19 – 10,000 | 10 – 2,137 | 45 | 0.44 | 2.47 | SFTK |
| Flat 3-bullet rosette | 5 | 20 – 10,000 | 10 – 1,882 | 45 | 0.2 | 2.43 | SFTK |
| **Aggregates:** | | | | | | | |
| **Pristine:** | | | | | | | |
| Evans snow aggregate | 1 | 32 – 11,755 | 50 – 2,506 | 35 | 0.20 | 2.39 | ESP |
| Tyynelä dendrite aggregate | 26 | 595 – 20,826 | 228 – 3,328 | 35 | 0.10 | 2.25 | ESP |
| 8-column aggregate | 8 | 19 – 9,714 | 10 – 5,000 | 39 | 65 | 3.00 | RSP |
| Small column aggregate | 17 | 105 – 3,855 | 37 – 738 | 35 | 0.14 | 2.45 | SFTK |
| Large column aggregate | 18 | 368 – 19,981 | 128 – 3,021 | 35 | 0.25 | 2.43 | SFTK |
| Small block aggregate | 21 | 100 – 7,328 | 72 – 1,665 | 35 | 0.21 | 2.33 | SFTK |
| Large block aggregate | 22 | 349 – 21,875 | 253 – 4,607 | 35 | 0.35 | 2.27 | SFTK |
| Small plate aggregate | 19 | 99 – 7,054 | 53 – 1,376 | 35 | 0.077 | 2.25 | SFTK |
| Large plate aggregate | 20 | 349 – 22,860 | 197 – 4,563 | 34 | 0.21 | 2.26 | SFTK |
| ICON hail | 30 | 120 – 5,349 | 94 – 5,000 | 35 | 380 | 2.99 | RC |
| ICON snow | 28 | 120 – 20,000 | 94 – 3,219 | 35 | 0.031 | 1.95 | RC |
| GEM hail | 29 | 120 – 5,031 | 94 – 5,000 | 35 | 540 | 3.02 | RC |
| GEM snow | 32 | 170 – 10,459 | 94 – 5,000 | 35 | 24 | 2.86 | RC |
| **Rimed:** | | | | | | | |
| Spherical graupel | 23 | 622 – 9,744 | 454 – 5,293 | 30 | 13 | 2.69 | SFTK |
| ICON graupel | 29 | 170 – 6,658 | 94 – 5,000 | 35 | 390 | 3.13 | RC |
| GEM graupel | 33 | 120 – 6,597 | 94 – 5,000 | 35 | 170 | 2.96 | RC |
| **Liquid:** | | | | | | | |
| Liquid sphere$^*$ | 25 | 1 – 50,000 | 1 – 50,000 | 200 | 523 | 3.00 | Mie |



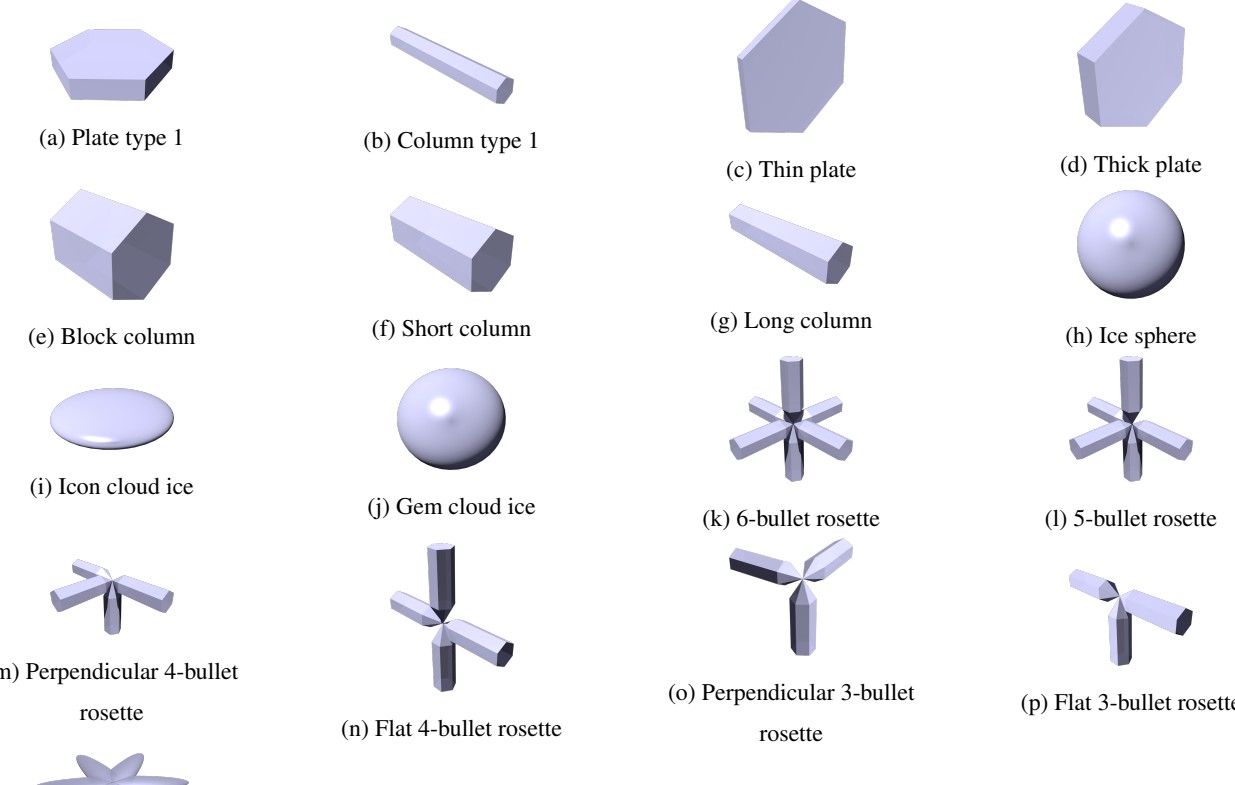

(a) Plate type 1  (b) Column type 1  (c) Thin plate  (d) Thick plate

(e) Block column  (f) Short column  (g) Long column  (h) Ice sphere

(i) Icon cloud ice  (j) Gem cloud ice  (k) 6-bullet rosette  (l) 5-bullet rosette

(m) Perpendicular 4-bullet rosette  (n) Flat 4-bullet rosette  (o) Perpendicular 3-bullet rosette  (p) Flat 3-bullet rosette

(q) Sector snowflake

**Figure 1.** Single crystal habits included in the first database version. Shown particle orientation varies between the habits.

**Table 2.** Database habit classification and structure.

| Habit classification | |
| --- | --- |
| Attribute | Possible cases |
| Orientation | Totally random |
| Phase | Ice |
| | Melting |
| | Liquid |
| Aggregation | Single crystal |
| | Aggregate |
| Riming | Pristine |
| | Rimed |



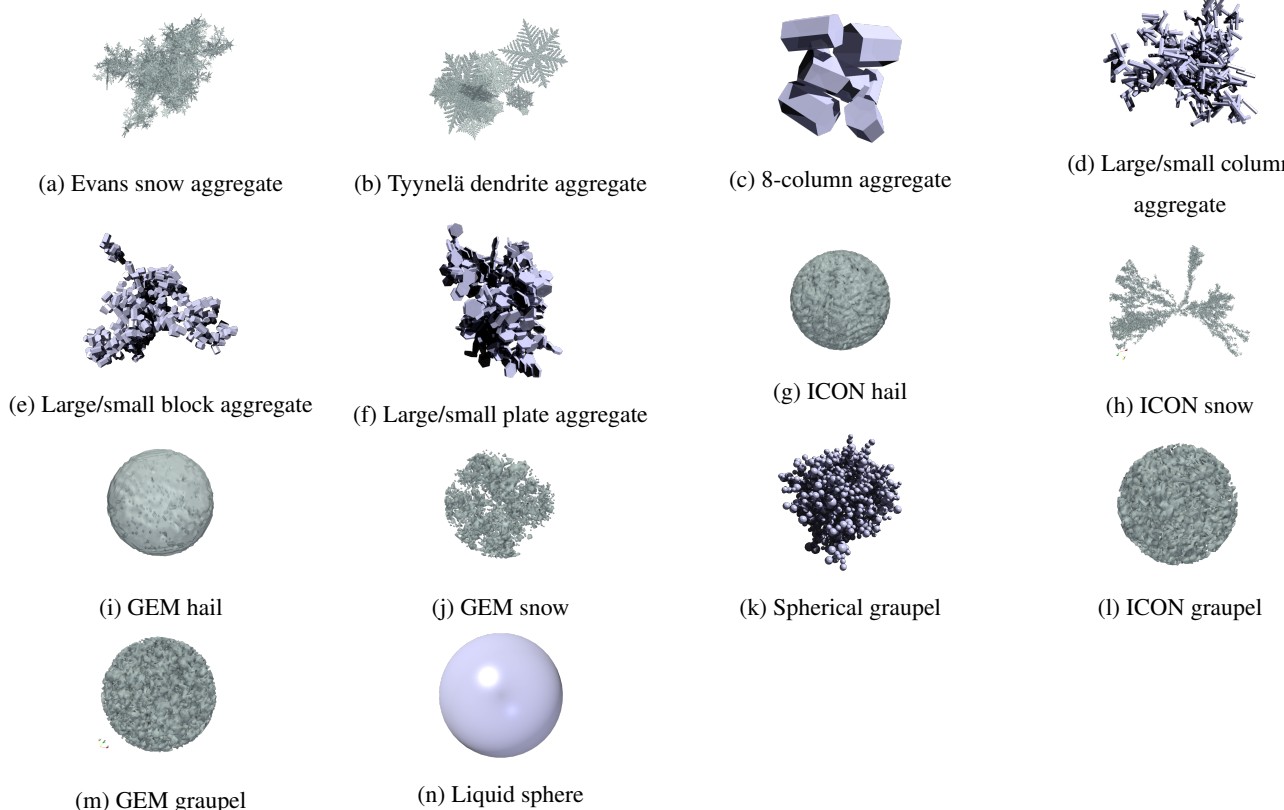

**Figure 2.** Aggregate and liquid habits included in the first database version. Shown particle orientation varies between the habits.

to give a reasonable coverage of particle sizes reported in literature. The smallest crystal of a habit is mostly $D_{\mathrm{veq}} = 10\,\mu\mathrm{m}$. The smallest aggregates mainly have $D_{\mathrm{veq}} < 100\,\mu\mathrm{m}$, but this criterion is not met when the constituting crystals are larger than this limit. The largest particle satisfies either the $D_{\mathrm{max}}$ or the $D_{\mathrm{veq}}$ limit, depending on $a$ and $b$ of the habit. The main exceptions, where neither the $D_{\mathrm{max}}$ and $D_{\mathrm{veq}}$ is reached, are aggregates consisting of relatively small crystals, where it was

5    computationally too costly to generate such large particles.

Habits consisting of solid spheres (handled by Mie code) were allowed to have both more sizes (200) and cover larger sizes ranges, in order to allow detailed reference calculations using these habits. The final size range of all habits is found in Table 1. The general strategy to select intermediate sizes is to aim for an approximately equidistant linear spacing up to $D_{\mathrm{veq}} = 100\,\mu\mathrm{m}$ and equidistant spacing in the logarithm of $D_{\mathrm{veq}}$ above.

10   Additionally, an upper limit with respect to size parameter ($x$) was introduced for practical reasons since the computational burden to calculate the scattering properties increases strongly with $x$. That it is possible to apply a limit on $x$ is due to the fact that the size range that contributes the most to bulk scattering properties is shifted towards smaller sizes when going up in frequency. That is, for a given (large) particle, $x$ itself increases with frequency, but the particle becomes less influential for the scattering properties of the overall local particle mix. This effect is illustrated in Fig. 3.

**Table 3.** Size range limits.

|  | Single crystals | | Aggregates | |
| --- | --- | --- | --- | --- |
|  | min | max | min | max |
| $D_{\max}$ | $10\,\mu\mathrm{m}$ | $10\,\mathrm{mm}$ | $100\,\mu\mathrm{m}$ | $20\,\mathrm{mm}$ |
| $D_{\mathrm{veq}}$ | $10\,\mu\mathrm{m}$ | $2\,\mathrm{mm}$ | $100\,\mu\mathrm{m}$ | $5\,\mathrm{mm}$ |
| $x$ | - | 10 | - | 10 |

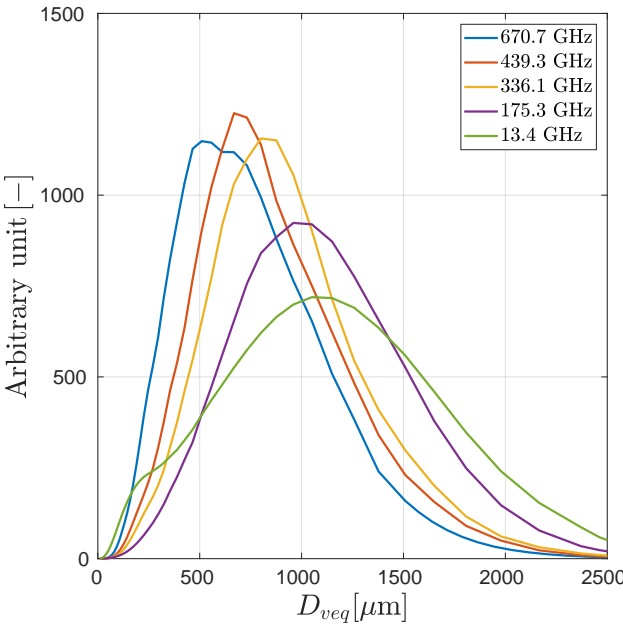

**Figure 3.** Contribution of differently sized particles to bulk extinction at a number of frequencies. The area under each curve has been normalised to unity. Calculated using the 6-bullet rosette habit and the tropical version of the Field et al. (2007) particle size distribution, assuming 270 K and an ice water content of $1\,\mathrm{gm}^{-3}$.

Fig. 4 demonstrates that $x = 10$ is a reasonable limit. It shows the estimated relative error in the calculation of bulk extinction assuming considering data to a given limit in $x$. Fig. 4 presents a worst case scenario as a relative high ice water content $(1\,\mathrm{gm}^{-3})$ and a frequency in the upper end of the database's coverage (671 GHz) are considered. Particles with $x < 1$ contribute marginally to the bulk extinction here indicated by the tail distribution staying close to 1 up to about $x = 1$. There are three
5   cases in Fig. 4, where the tail distribution is well above 10% even at $x = 10$. These cases are all combinations between the PSD of Field et al. (2007) and habits having $b = 3$. This PSD is intended for snow type hydrometeors, where $b$ around 2 is expected (Baran, 2012b). That is, these three cases of concern represent fairly unrealistic, or extreme, assumptions.





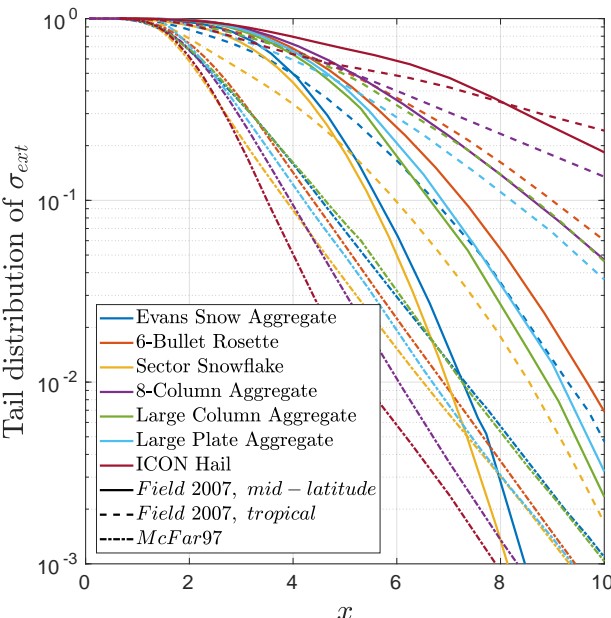

**Figure 4.** Tail distribution of extinction (see text for details), at 670.7 GHz and assuming 270 K and an ice water content of $1\,\mathrm{g\,m^{-3}}$, for some combinations of habit and particle size distribution (PSD). Solid and dashed lines are results using the tropical and mid-latitude version of Field et al. (2007) as PSD, while dashed-dotted lines show results using the PSD of McFarquhar and Heymsfield (1997).

It is not unlikely that further problematic situations with respect to the PSD exist apart from those three cases presented above, and occasional errors above 10% can not be ruled out. On the other hand, it should be remembered that the water vapour absorption at 671 GHz is high, giving a decreased sensitivity to lower altitudes, where most large particles are found most frequently, when observed by satellite instruments. In summary, our interpretation of Fig. 4 is that the error by applying

5    a limit at $x = 10$ should in general be well below 10%.

The size limit with respect to $x$ is evaluated separately for each frequency. This means that at low frequencies data were calculated for all sizes, while data can be lacking for large particles at higher frequencies. The limit was applied in such way that, if possible, coverage up to $x = 10$ is maintained. That is, assuming that particles matching $x \geq 10$ are available, the smallest one exceeding the limit still got included.

10    **2.6    Frequencies and temperatures**

The overall frequency coverage was set to be 1 - 886.4 GHz. Inside this range, optical properties are given for 34 frequencies (Table 4). A majority of the frequencies were selected to bracket channel sets found on both existing and planned operational microwave sensors. These are in the following termed "channels" with arbitrary channel numbers assigned (see (Table 4). For example, data are provided for 50.1 and 57.6 GHz, forming channel 3, which should provide coverage of the standard set





of microwave temperature sounding channels. Additional frequencies were hand-picked to cover standard radar wavelengths, as well as to obtain a general coverage between 1 and 15 GHz. Above 200 GHz, the frequencies are mainly governed by the channels of the planned Ice Cloud Imager (ICI) mission (Kangas et al., 2012). The upper end at 886.4 GHz was set in order to support the ISMAR (Fox et al., 2017) and IceCube (Wu et al., 2015) instruments.

It is stressed that the database should be used with high care for frequencies outside the defined channel ranges. Whether interpolation between the channels can be applied must be judged from case to case. For example, interpolation to 170 GHz should not be problematic as the distance between channel 6 and 7 is small (in fact smaller than the width of channel 7 itself), while frequencies around e.g. 550 and 770 GHz can not be claimed to be properly covered by the database. The relevant measure to judge if an interpolation can be performed should be the ratio between spacing and frequency, not the absolute size of the

frequency spacing. As a general rule, only the closest database frequencies should be used when performing interpolation.

**Table 4.** Included database frequencies. Frequencies above 18.6 GHz organised into channels (see text).

| Channel | 1 | 2 | 3 | 4 | 5 | 6 | 7 | 8 | 9 | 10 | 11 | 12 |
|---|---|---|---|---|---|---|---|---|---|---|---|---|
| Freq. | 18.6 | 31.3 | 50.1 | 88.8 | 115.3 | 164.1 | 175.3 | 228.0 | 314.2 | 439.3 | 657.3 | 862.4 |
| (GHz) | 24.0 | 31.5 | 57.6 | 94.1 | 122.2 | 166.9 | 191.3 | 247.2 | 336.1 | 456.7 | 670.7 | 886.4 |
| Other frequencies: | | | | | | | | | | | | |
| 1, 1.4, 3, 5, 7, 8, 9, 10, 10.65, 13.4, 15 | | | | | | | | | | | | |

Optical properties are given for multiple temperatures, rather than a set of real and imaginary refractive index values. The reason for this is that the used parameterisations, Mätzler (2006) and Ellison (2007) for ice and water, respectively, should predict the refractive index with sufficiently high accuracy (Sec. 4.1). The selected set of temperatures are 190, 230 and 270 K for hydrometeors consisting solely of ice. For pure liquid droplets, the temperatures are 230, 250, 270, 290 and 310 K. See

Sec. 4.1 for further details regarding issues related to refractive index and to temperature interpolation.

## 3   Software and processing

This section provides an overview of the software used for the generation of the data. The generation of the database consists of two main steps. First, a description of the particle shape must be acquired, either from some analytic expression or by physical models of varying complexity (Sec. 3.1). Second, the actual single scattering properties are calculated (Sec. 3.2), mainly by

the Discrete Dipole Approximation (DDA) method (Sec. 3.2.1). The DDA calculations are complemented with a set of quality checks and modifications, with the purpose of ensuring high quality and consistency.



### 3.1 Shape data generation

#### 3.1.1 Snowflake toolkit

Particle shape data have mainly been generated using an internally developed software (Rathsman, 2016)[1], called the *Snowflake Toolkit* (SFTK). The toolkit's main features are a format for representing crystal shapes in an analytic way, an algorithm for
simulation of aggregation, and sampling routines for generating DDA input shape data.

The toolkit's shape format is used both internally in the simulation software and to specify output data. The format represents particles as polygon meshes. This makes it straightforward to not only represent simpler shapes, such as hexagonals and bullets, but also aggregates of said crystals. Further, the same shape data file can be used both as input for dipole sampling or aggregate simulation. The dipole sampling routines can take both crystals or aggregates as input. An advantage of using a polygon
representation along with a sampling routine is that the grid resolution can be set arbitrarily, depending on application. The grid accuracy can thus be adapted to frequency.

As mentioned, the software includes capabilities to simulate aggregation. A semi-physical model, inspired by Maruyama et al. (2005), is applied. The model can be considered to describe a small section in a cloud, with a number of ice particles inside the volume. Several events can occur at each iteration: the aggregation of two particles, growth of a new crystal, particle
melting or fallout of the cloud. The likelihood of these events depends on a number of factors such as size, estimated fall speed and control parameters. Contrary to most aggregate models used in the field, this model allows for the aggregation of two aggregates; the algorithm can consider aggregation between any particles that can be represented by the software's shape format.

Aggregation is modelled as if the two particles collide with each other at random angles and centre off-sets. Aggregate
sticking is only allowed to occur at particle faces, where the two involved surface normals are forced to be be parallel (i.e. face to face sticking is ensured). For each candidate aggregation, it is checked if it has created any overlap between any combination of building blocks. If any overlap is found, the aggregation is discarded or a new try is made depending on the exact settings. There are a number of available control parameters. One such parameter is the number of particles allowed to populate the cloud at a given iteration. The available memory has to be taken into consideration here, but at least 5,000 particles can be
handled by a standard desktop computer. Growth, melting and drop-out ratio parameters are used to determine the relative likelihood of the different possible events.

As a complement, also the aggregation of spheres is handled. This version is more straightforward, but less based on physics. The main application should be to generate heavily rimed particles. The model describes a single aggregate particle, originally only a single sphere, on to which new spheres are added. Each new sphere is inserted from a random angle. It is allowed to let
the spheres "tunnel" and reach the interior of the particle. The tunnelling feature is required to reach higher effective densities, but is not obligatory. It is also possible to set the angular distribution of the incoming spheres to be non-uniform, as a way of influencing the shape of the resulting aggregate. Other options and features available include: maximum diameter (constraining the growth of the particle) and fill ratio (ensures that a minimum effective density is achieved).

---

[1]Code available at https://github.com/milasudril/snowflake-toolkit





### 3.1.2 RimeCraft

There is frequently a desire to generate particles fulfilling Eq. 5 for some pre-defined $a$ and $b$. This task is hard to tackle with the Snowflake toolkit, and, as an alternative, a set of Matlab functions was developed for this task. As the function set should also be useful for adding different degrees of riming, it was named *RimeCraft* (RC). It works by adding 3-d cubic blocks

directly on the shape grid. Different rules can be defined for how to add blocks. Eq. (5) is fulfilled by growing the particle inside subsequent thin layers, and calculating a fill ratio for each layer from the target $a$ and $b$. The position of a new dipole is selected randomly, but weights can be specified to tune how a new dipole prefers to have "neighbours" or not. This controls the overall granularity of the particle. Finer details of the structure can be modified by setting different weights (that can be both positive and negative) with respect to face, edge and vertex neighbours. A complete habit can be created either by taking

a snapshot at each intermediate size, as a single particle grows, or by rerunning RimeCraft for each target $D_{\mathrm{max}}$.

An additional feature (not yet used for the database), is to simulate the melting of a particle. This feature largely follows the algorithm of Johnson et al. (2016). Here an existing, gridded, particle is used as starting point. The algorithm consist of two main steps at each iteration. First, all ice grid points are assigned weights based on the radius from the centre (to take the temperature gradient into account) and their number of icy neighbours. A portion of the grid points with the highest weights

are selected and converted to liquid. The liquid grid points are then allowed to move, in order to mimic surface tension. The iterations continue until the desired melting fraction is achieved. A column particle, for example, would begin to melt at its tips, successively melt inwards and be converted to a drop.

### 3.2 Calculation of optical properties

### 3.2.1 ADDA

A vast majority of the scattering data has been produced by the Amsterdam DDA (ADDA), a DDA implementation developed by Yurkin and Hoekstra (2011). The basic idea of the DDA method is to represent the particle by a discretised equidistant Cartesian grid, where each grid point represents an electric dipole with polarisation $\mathbf{P}$. The DDA solution can be formulated as

$$\alpha_i^{-1}\mathbf{P}_i - \sum_{j \neq i}\overline{\mathbf{H}}_{ij}\mathbf{P}_j = \mathbf{E}_i^{inc}, \tag{6}$$

where $\alpha$ is the dipole polarisability, $i$ and $j$ are grid point indices, $\overline{\mathbf{H}}$ the total interaction term, and $\mathbf{E}^{inc}$ the incoming electric field (Yurkin and Hoekstra, 2011). The equation is solved iteratively, and output quantities are derived via the resulting total electric field $\mathbf{E}_i$. This field can be derived when the polarisation field is known, using $\mathbf{P}_i = V_{\mathrm{dp}}\chi_i\mathbf{E}_i$, where $\chi_i$ is the susceptibility of the medium and $V_{\mathrm{dp}}$ the volume of a dipole. Without going into details, there are different formulations known for the susceptibility and the interaction terms. For this database, the "lattice dispersion relation" was used for the susceptibility and

"interaction of point dipoles" for the interaction term (Draine and Flatau, 1994). The ADDA calculations are stopped when the relative norm of the residual, essentially the relative difference between the right and left part of Eq. (6), reaches a user




specified limit, denoted as $\epsilon_{\mathrm{DDA}}$. The selected value of $\epsilon_{\mathrm{DDA}}$ thus determines the accuracy of the output and we have applied

$$\epsilon_{\mathrm{DDA}} = 10^{-2}, \tag{7}$$

aiming for an accuracy in the order of a few percent.

The shape data, in the form of shape files, are the most important input required by ADDA. The discretised grid of a particle
is defined in the shape file as three columns of grid coordinates. The grid is scaled such that only integer values are used as
coordinates. An extra column can be added to index points with different refractive indices, i.e. only required for heterogeneous
particles. Other essential input parameters are the particle volume equivalent radius ($r_{veq}$), radiation wavelength ($\lambda$), number of
dipoles per wavelength ($dpl$), propagation vector of incoming radiation and the scattering grid, and orientation of the particle
(Euler angles using zyz-notation). The variable $dpl$ is critical as it must be sufficiently large. We determine $dpl$ such that we
fulfil the standard criterion:

$$|m|kd < 0.5, \tag{8}$$

where $|m|$ is the complex refractive index, $k$ the wavenumber, and $d$ the dipole size. The microwave refractive index of ice
results in that $dpl$ should be $\approx 22$ or higher. As an additional rule, the minimum number of dipoles was set to 1000, to enable
that also the shape of particles of small $x$ is resolved in a reasonable manner.

ADDA handles two orientation cases, either fixed or totally random. The later case is handled by Romberg integration (Davis
and Rabinowitz, 1984). This method starts out with two scattering calculations, at orientation angles spaced 180° apart. The
integrated average is then updated iteratively by adding scattering calculations at new angles, until convergence is achieved.
This tends to happen faster for calculations of small size parameter $x$, whose scattering functions are less varying compared to
large $x$. Note that while this averaging is expressed in Eq. (2) by three angles, only $\beta$ and $\gamma$ are varied in ADDA, as rotation
over $\alpha$ is equivalent to rotation of the polarisation plane, which can be performed analytically. By effectively removing one
dimension from the orientation "grid", significant calculation time is saved. The convergence criterion refers here to maximum
residual error of the averaging over each Euler angle, $\epsilon_{\mathrm{avg}}$. We have used

$$\epsilon_{\mathrm{avg}} = 10^{-2} \tag{9}$$

for both Euler angles. All ADDA calculations found in the database so far have been calculated using ADDA's orientation
averaging.

ADDA can output the absorption cross-section $\sigma_{\mathrm{a}}$, extinction cross-section $\sigma_{\mathrm{e}}$, Mueller matrix $\mathbf{M}$, and amplitude matrix $\mathbf{S}$.
The Mueller matrix is a 4-by-4 matrix that relates the Stokes vector of incoming and scattered radiation, and is defined with
respect to the scattering reference frame. The scattering cross-section is related to the Mueller matrix by

$$\sigma_{\mathrm{s}} = \frac{1}{k^2} \int_{4\pi} \mathbf{M}_{11} \, \mathrm{d}\boldsymbol{n}', \tag{10}$$

where $k$ is the wavenumber. The amplitude matrix relates also incoming and scattered radiation, but operates on the electric
field. Both the Mueller and amplitude matrices have incoming and scattered angular dependence. Note that it is not possible



to compute the amplitude matrix in conjunction with ADDA's orientation averaging option, as the amplitude matrix refers to coherent radiation and the averaging assumes incoherent interaction between the particles.

Some sanity checks, and possibly also some smaller corrections, are applied on the ADDA output:

1. Positivity: The cross-sections $\sigma_e$ and $\sigma_a$ must be positive ($\geq 0$). The same applies to the $\mathbf{M}_{11}$ element at all angles. However, a violation of 1 % relative to the total extinction is allowed for $\sigma_a$, in which case $\sigma_a$ is re-set to

$$\sigma_a = \sigma_e - \sigma_s. \tag{11}$$

   More negative $\sigma_a$ is not tolerated.

2. Consistency: The extinction must be larger than both the scattering and absorption separately, i.e. $\sigma_e \geq \sigma_a$ and $\sigma_e \geq \sigma_s$. An violation of 1 % relative to $\sigma_e$ itself is allowed.

3. Energy conservation: The extinction must equal the absorption and scattering cross-section, i.e. $\sigma_e = \sigma_a + \sigma_s$. At this point, differences up to 30% have been allowed. This is an unsatisfactorily situation, but issues around the sector snowflake habit forced us to apply this high value. These issues are likely associated with the extreme aspect ratios of large sector snowflakes, and we aim for having a more demanding limit in the next database version. The maximum deviation found among other habits is $< 5\%$.

DDA calculations that do not fulfil all of above criteria are recalculated with higher accuracy criteria.

Finally, the database quantities $\mathbf{K}$, $\mathbf{a}$ and $\mathbf{Z}$ are set as

$$\mathbf{a}_1 = \sigma_a, \tag{12}$$
$$\mathbf{Z} = \frac{1}{k^2}\mathbf{M}, \tag{13}$$
$$\mathbf{K}_{ii} = \sigma_e = \sigma_a + \sigma_s, \tag{14}$$

Other elements in $\mathbf{a}$ and $\mathbf{K}$, i.e. $i \neq 1$ in $\mathbf{a}_i$ and off-diagonal elements in $\mathbf{K}$, are set to zero according to the theory for totally randomly oriented particles (Mishchenko et al., 2002).

### 3.2.2 Mie

Two habits represented by spheres are included in the database, liquid and solid ice spheres. For the derivation of these scattering properties it is appropriate to make use of the well-established Mie theory, which provides an exact solution and is much more computationally efficient than DDA. The implementation in MATLAB by Mätzler (2002) was used. Two functions are used, `mie` and `mie_12`, where the former calculates scalar quantities, such as scattering efficiencies. The efficiency parameters are normalised scattering quantities with respect to the particle cross-sectional area, for example in the case of extinction:

$$\sigma_e = \pi r^2 Q_e, \tag{15}$$



where $r$ is the sphere radius. `mie_12` calculates the scattering amplitude matrix elements $S_{11}$ and $S_{22}$. These elements describe the angular dependence of the plane-perpendicular and plane-parallel components of the scattered light relative to the incoming light ($S_{12}$ element is zero, since spheres cause no depolarisation). The amplitude matrix is converted to scattering matrix $\mathbf{Z}$ using the wrapper code provided by the Atmlab[2] package.

## 4 Microphysical and dielectric data

This section deals with the microphysical and dielectric data employed for the database. Specifically, Sec. 4.1 discusses the refractive index models used, Sec. 4.2 and Sec. 4.3 describe single crystal and aggregate shape data, respectively, Sec. 4.4 explains how rime is included, and the subject of Sec. 4.5 is liquid particles.

### 4.1 Refractive index

Beside size and shape of particles, single scattering properties are determined by the dielectric properties of the particle material. These are commonly described by the complex refractive index, where the real part mainly affects the scattering properties while the imaginary part primarily determines the absorption. The refractive index of liquid and frozen water at microwave frequencies varies significantly with frequency as well as with temperature.

A review of (water) ice refractive index models is shown in Eriksson et al. (2015), and a comparison between some models is also found in Fig. 5. The refractive index model by Ray (1972) is included in Fig. 5 as it strongly deviates from more recent parameterisations, based on a more wide set of laboratory measurements, but it has still been applied in some recent studies (Lu et al., 2016; Gong and Wu, 2017). The most significant deviations are found for the imaginary part. This causes concern for at least passive measurements, as ice absorption has been shown to play a non-negligible role for such observations. (e.g. Eriksson et al., 2011b, 2015; Ding et al., 2017).

In Eriksson et al. (2015), the ice refractive index model by Mätzler (2006) was suggested as a good choice for the full microwave region (up to about 1 THz) and Earth atmospheric temperatures. Newer models (e.g. Warren and Brandt, 2008; Iwabuchi and Yang, 2011) refer to, or are essentially identical, to Mätzler (2006), i.e. the recommendation remains valid. Hence, for our database, we set the refractive index of ice according to Mätzler (2006).

For liquid water, the model of Ellison (2007) is applied, which provides refractive indices for frequencies up to 25 THz and temperatures between 0°C and 100° based on empirical fits to an extensive collection of water refractive index data (Ellison et al., 1996). Deviations to the model by Turner et al. (2016), which focuses on super-cooled liquid water but does not cover sub-millimetre frequencies, remain smaller than 30% even at the lowest temperatures of concern (-40°C). Hence, although Ellison (2007) is, in the strict sense, not recommended for super-cooled liquid water, it is applied here at all temperatures.

Finally, a smaller study on the required temperature grid for ice habits was performed. Assuming optical properties of solid Mie spheres, a number of forward calculations with varying temperature grid resolution were compared. Forward calculations using a 2.5 K spaced temperature grid covering 150 to 270 K were used as reference. A number of coarser grids, all equally

---
[2]http://www.radiativetransfer.org/tools/




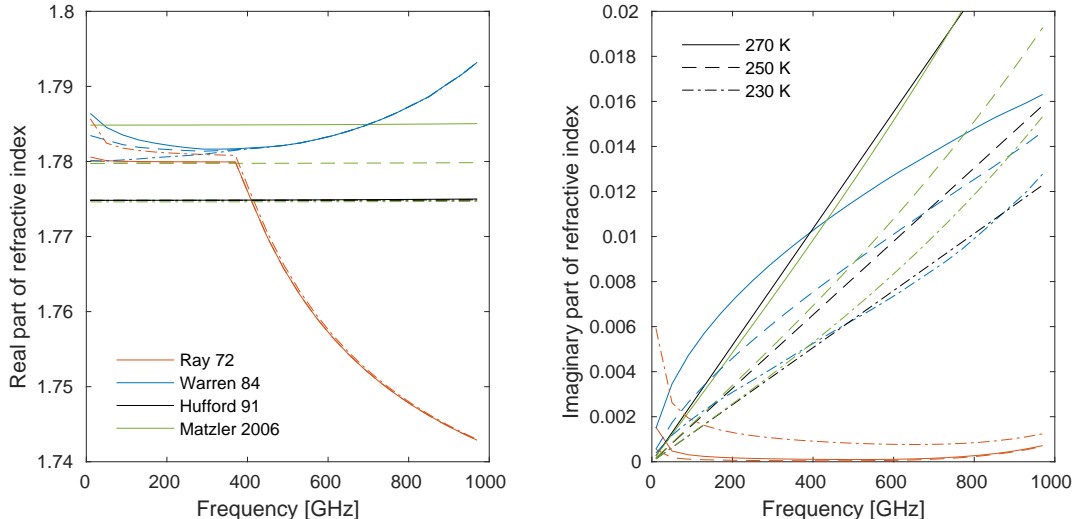

**Figure 5.** Real (left) and imaginary (right) part of the refractive index of pure ice, for selected temperatures as a function of frequency, according to Ray (1972), Warren (1984), Hufford (1991), and Mätzler (2006).

spaced between 190 to 270 K and applying linear inter- and extrapolation in temperature, were compared to the reference set. Simulations were performed for all channels employed by the airborne radiometer ISMAR (Fox et al., 2017) and 5000 atmospheric scenarios contained in Eresmaa and McNally (2014), in order to maximise the temperature profile diversity. Deviation statistics over these 5000 cases and the 13 channels were derived. Fig. 6 summarises the results, showing that for

a setup with three temperature grid points, maximum deviations to the reference simulations over all scenarios and channels are $< 2$ K, and the 99.7-percentiles of differences are $< 0.2$ K. For five grid points, maximum deviations fall below 0.3 K and the 99.7-percentile below 0.05 K. The accuracy can be improved further by applying higher order polynomial temperature interpolation. The three point grid was judged as a good compromise between attainable simulation accuracy and the calculation burden required to generate the database.

**4.2  Ice crystals shapes**

The single crystal habits provided by the database are summarised in the upper part of Table 1. Example visualisations are found in Fig. 1. The crystal shapes are mainly based on parameterisations taken from the literature (Liu, 2008; Hong et al., 2009), but also some new crystal habits have been defined. For completeness and as a reference, also the ice sphere habit was included.

For crystal habits, external data are restricted to numerical parameterisations of shape properties, while the actual shape data matching those parameterisation were generated by the Snowflake toolkit. The long column, short column, block columns, thick plates and thin plate, taken from Liu (2008), are all hexagonal and have axial ratios of 4, 2, 1, 0.2 and 0.05, respectively. These axial ratios were defined by Liu (2008) to be constant over all sizes. As a consequence, they all obtain $b = 3$. In contrast, plate





**Figure 6.** Error statistics of ISMAR observation simulations, depending on temperature grid length used to represent ice hydrometeor scattering properties.

and column type 1, with dimensions taken from Hong (2007), are defined to have a $b$ of 2.48 and 2.05, respectively, implying an axial ratio that grows with mass. These definitions were based on measurements of ice crystal dimensions (Auer Jr and Veal, 1970; Mitchell et al., 1994; Yang et al., 2000). The sector snowflake, defined in Liu (2008), is an idealised representation of a snowflake, consisting of three intersecting ellipsoids with their mass centres positioned at the origin. This parameterisation

was based on surface measurements of aggregates (Kajikawa, 1982).

The different bullet rosette habits all consist of hexagonal bullets connected at the tips. The crystal dimensions are taken from Hong (2007), based on measurements listed in that paper. All bullets grow in a similar fashion, in the sense that all bullets with equal mass will have the same dimensions regardless of the considered habit. Furthermore, in all versions, the bullets are aligned perpendicular to each other. The flat 3- and 4-bullet rosettes have all their arms lying in a common plane. The

perpendicular versions have one arm rotated 90°, resulting in more compact particles.

Two crystal habits were generated specifically to match the $a$ and $b$ assumed for cloud ice in two atmospheric models: GEM (Global Environmental Multiscale Model) and ICON (Icosahedral non-hydrostatic general circulation model). The GEM model was prepared for the EarthCARE mission (Côté et al., 1998) and the ICON model is a part of the German HD(CP)2 project[3]. The GEM values ($a = 440$ and $b = 3$) imply a habit with low and constant aspect ratio, and are in fact very close to the

ones of solid spheres. Accordingly, it was decided to apply spheroids for GEM cloud ice. For simplicity and to have some additional reference spheroidal reference data, the same basic shape was selected for ICON cloud ice. The scattering properties of spheroids can be calculated efficiently by the T-matrix method, but for consistency reasons and to avoid possible numerical problems of T-matrix at higher size parameters (Mishchenko and Travis, 1998), also these habits were processed using the Snowflake toolkit and ADDA.

---

[3]http://hdcp2.eu



### 4.3 Ice aggregate shapes

The ARTS database includes habits both developed specifically for this database and from third party sources. For calculations at higher frequencies, aggregates are arguably lacking in the current scattering databases and the aim has been to include a relatively diverse set of aggregate habits.

### 4.3.1 Third party data

Aggregate third party data come in two forms, either in the form of shape data ready to be used in the DDA calculations, or as shape descriptions taken from literature. The 8-column aggregate, originally defined by Yang and Liou (1998) and also represented in Hong et al. (2009), is of the later case. It is constructed of eight columns of varying sizes. The aspect ratios and relative positions of the columns do not change with mass, and the habit has a $b = 3$. The $a$-value corresponds to an effective density of 13.5 %.

The Evans snow aggregate (Evans et al., 2012) and Tyynelä snowflake aggregate (Tyynelä, 2011) are both habits where explicit third party shape data have been used. They both have features common with the aggregates generated by the Snowflake toolkit, in that they are generated in a stochastic manner, with multiple particles kept track off at the same time. As the Snowflake toolkit, the generation of snow aggregate in Evans et al. (2012) was based on Maruyama et al. (2005). The dendrites were randomly oriented during aggregation events, and they were connected with an overlap of 5 % in volume. The Tyynelä aggregates were generated assuming a quasi-horizontal alignment for the dendrites, more precisely a Gaussian orientation distribution with a standard deviation of 2°. The publically available shape data are of too low resolution for large parts of the database. Higher resolution data was obtained by correspondence with J. Tyynelä and applied here for these aggregates. The shape data by Tyynelä and Evans have both a somewhat poor resolution ($dpl = 15$ at 664 GHz) and corresponding scattering data have likely limited accuracy above roughly 460 GHz.

### 4.3.2 Database specific

The aggregate simulator of the Snowflake tool-kit (Sec. 3.1.1) was used to create a number of aggregate habits. Throughout, hexagonals have been used as constituent crystals. Separate simulations were performed using mean axis ratios of 5 for columns, 1.25 for blocks (short columns), and 1/6 for plates. Furthermore, dimensions of each constituent crystal, length $L$ and and side length $a$ (Sec. 2.3), are not set with an exact fixed value, but are each selected randomly from Gamma distributions, using a standard variation of 25 %. Since $L$ and $a$ are selected independently, the aspect ratios are randomly distributed as well, around the specified mean value. Separate simulations were also run for different constituent crystal mean maximum dimension $D_{\mathrm{max}}$, for which two values, 100 and 350 $\mu$m were chosen. The motivation for having two constituent crystal sizes is that the simulations could not provide large enough particles when using 100 $\mu$m crystals. Hence, habit versions with 350 $\mu$m crystals were created as well. In summary, six simulations in total were performed, using three aspect ratios and two crystal maximum dimensions.



The details of this set are summarised in Table 5, and example figures of the particles are shown in Fig. 2. A difference to the Evans aggregate is that no overlap between crystals is allowed, and the crystals are always connected with their faces against each other. Since each simulation generates thousands of particles, a filtering had to be applied. In line with the statement in Sec. 2.4, that habits should follow a single mass-size power-law, we fit Eq. (5) to the whole set of particles and make a
somewhat ad hoc selection among the particles that best match the derived $a$ and $b$ to obtain a logarithmically spaced grid in $D_{\mathrm{veq}}$.

**Table 5.** Overview of aggregates produced using the Snowflake toolkit. The columns axial ratio and $D_{\mathrm{max}}$ refer to mean values of constituting crystals, while the last two columns refer to complete aggregates.

|                       |     | Axial ratio | Crystal              |       |      |
| --------------------- | --- | ----------- | -------------------- | ----- | ---- |
| Habit name            | Id  | L/(2a)      | $D_{\mathrm{max}}$ [$\mu$m] | $a$   | $b$  |
| Small Column Aggregate | 17  | 5           | 100                  | 0.14  | 2.45 |
| Large Column Aggregate | 18  | 5           | 350                  | 0.25  | 2.43 |
| Small Block Aggregate  | 21  | 1.25        | 100                  | 0.21  | 2.33 |
| Large Block Aggregate  | 22  | 1.25        | 350                  | 0.35  | 2.27 |
| Small Plate Aggregate  | 19  | 1/6         | 100                  | 0.077 | 2.25 |
| Large Plate aggregate  | 20  | 1/6         | 350                  | 0.21  | 2.26 |

Furthermore, habits matching snow and hail hydrometeor assumptions in the GEM and ICON models were generated using the RimeCraft function set. The hail habits were initiated with a single cubic "block" and were allowed to grow inside a spherical volume. Together with the $a$ and $b$ of the GEM and ICON hail habits, this resulted in (macroscopically) quasi-
spherical particles exhibiting some surface roughness and internal hollows. For the snow habits, some initial structure was needed to create a visual resemblance of snowflakes. Small 3d- and 2d-crosses with one block thick arms were used for GEM and ICON habits, respectively. With $b \approx 3$, GEM snow-like hail was allowed to grow in a spherical volume, while for the ICON snow habit with $b \approx 2$ the allowed volume for growth was set to be plate-shaped.

## 4.4    Rimed particles

Three different habits designed to represent graupel are included in the database. All three can be considered as heavily rimed particles, i.e. only composed of rime. The spherical graupel was generated using the Snowflake toolkit module treating aggregation of spheres. The angle distribution of the incoming spheres was set to be isotropic, resulting in an overall spherical shape of these graupel particles. Furthermore, the constituting spheres have randomly distributed diameters, selected from a gamma distribution with a mean radius of 300 $\mu$m, and a standard deviation of 75 $\mu$m. Furthermore, the habit was designed to
have an effective density of approximately 15% in order to roughly match observations (Mitchell, 1996).

The ICON and GEM graupel habits were generated in the same general way as the ICON and GEM hails habits, but the settings were tuned to create a granular pattern when examining slices through the resulting graupels. Particles that can be classified as lightly or moderately rimed can be generated by RimeCraft, and are intended to be included in a next database version.





### 4.5 Rain and liquid cloud drops

The liquid phase is so far limited to spherical drops, with scattering properties calculated by Mie theory (Sec. 3.2.2). The number of sizes was set to 200, and, as mentioned, data are provided for five temperatures between 230 and 310 K. Spherical drops should be an acceptable approximation for liquid cloud and drizzle particles, but not for larger rain drops, and liquid

drops of non-spherical shape should be part of the next database version.

## 5 Results

This section presents example characteristics and results of the database. The ambition is to give a basic overview of the database. Exploration of all details of the scattering data provided is left for other publications and the users of the database.

### 5.1 Comparisons

A number of comparisons have been made in order to obtain a rough assessment of the accuracy of the data. At an early stage, dedicated tests were made to ensure that there is an acceptable agreement between our usage of ADDA and equivalent results obtained by the T-matrix implementation of Mishchenko and Travis (1998). Some deviations were unavoidable as T-matrix is a fully analytical method, while DDA is numerical and operates with discretised shape data. This is of special concern for particles where the properties vary quickly as a function of size (due to "Mie resonance features"). DDA can shift the pattern

somewhat in size, causing the data for a specific size to deviate signifantly. However, this should not be critical as these details are largely removed when averaging over size to obtain bulk properties. Regarding actual content of the database, this feature is present in the ICON and GEM habits, consisting of spheroids.

  Further, Fig. 7 compares the extinction of some hexagonally shaped particles with T-matrix calculations for cylinders. The T-matrix cylinders were given the same length/height and same cross-sectional area, i.e. volume (and hence also mass) is

maintained. The ADDA results for back-scattering mainly agree with T-matrix, but show also indications of some instability. A closer examination of the poorer cases revealed that high deviation in back-scattering is associated with low number of orientations used by ADDA, indicating that its iteration criterion is not optimal with respect to this quantity. That is, the Romberg scheme has stopped prematurely with respect to back-scattering. On the other hand, an acceptable accuracy appears to have been reached for extinction. As expected, the extinction deviates between ADDA and T-matrix, but not strongly, the

difference is throughout below 4%. These results give additional indications on the accuracy of the database, but also indicate that the microwave extinction of ice particles primarily follows overall particle properties, such as mass and maximum size, while surface features, such as roughness, have a relatively low impact. That is, if plates and columns are treated to be hexagonal or cylindric has only a marginal impact on microwave optical properties.

  For habits, where external shape definitions have been used, comparison with their source databases (Liu, 2008; Hong

et al., 2009) is possible. However, these comparisons involve interpolation in temperature, frequency and size, and again a perfect agreement can not be expected. In order to minimise the impact of these interpolations, close matches in frequency





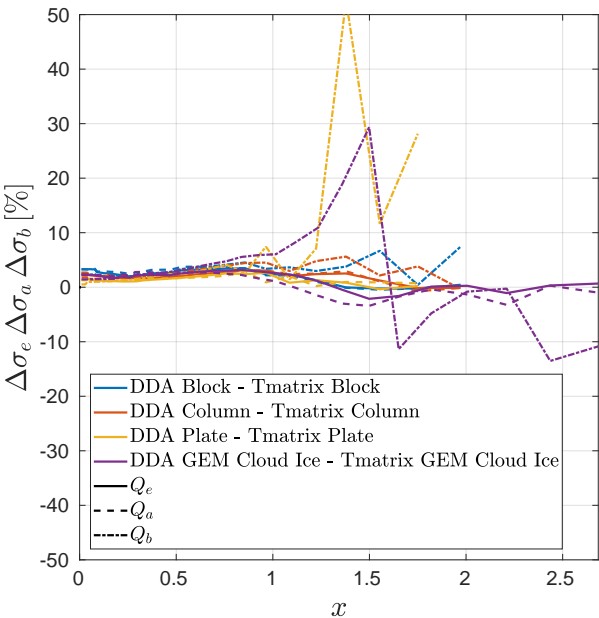

**Figure 7.** Comparison between the extinction, absorption and back-scattering of short columns, block columns and thick plates in the database, and T-matrix results using cylinders having the same length and mass. Shown as a function of size parameter $D_{\mathrm{veq}}$, with relative difference calculated as $(\sigma - \sigma_{\mathrm{Tmatrix}})/\sigma_{\mathrm{Tmatrix}}$. Temperature is 230 K and frequency is 94.1 GHz.

and temperature between the databases have been selected. Discrepancies will also depend on differences in DDA convergence criteria selected as well as on the exact sampling methodology used when discretising the particle shapes. Of concern is also that Hong et al. (2009) set dipole sizes following our Eq. (8), while Liu (2008) used a more strict criterion resulting in a factor of two smaller dipole size. Both performed DDA calculations for a set of specific particle orientations and based the orientation averaging on these data, but the number of orientation angles differ. Tests to estimate the impact of these later factors are reported by both Liu (2008) and Hong et al. (2009).

As we don't have access to the discretised shape data used in other databases it is not possible to assess applied particle shapes on that level, but reported masses can be compared. The masses obtained by us show insignificant deviations to the corresponding values reported by (Liu, 2008) and Hong et al. (2009), with the exception of the sector snowflakes. For this habit, discrepancies in mass up to 9% are found (Fig. 8). A simple discretisation of the particle shape normally results in a shift of the particle mass. Our strategy has been to apply a rescaling of the discrete data to ensure that the final data corresponds to the target mass. Already Geer and Baordo (2014) noted that the sector snowflake habit in Liu (2008) shows an unsmooth variation in mass, and it seems that no similar rescaling was applied by Liu (2008). The discretisation is a particularly critical issue for the sector snowflake habit, as these particles have the highest aspect ratios and can be only a few "dipoles" thick. Just



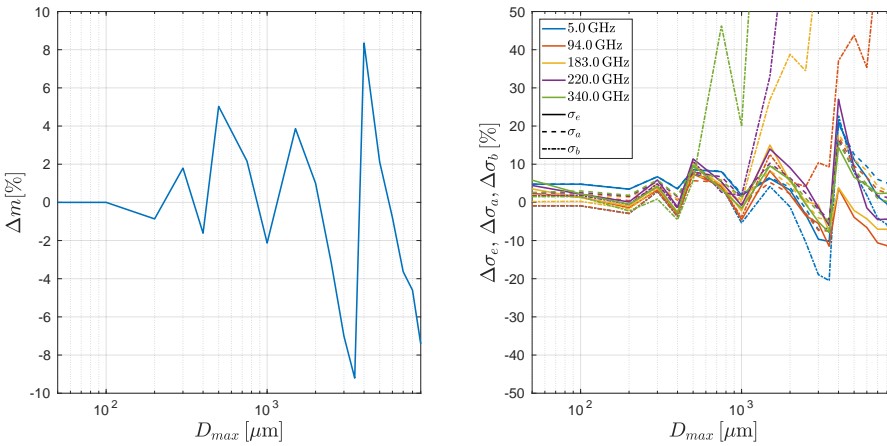

**Figure 8.** Comparison of sector snowflake data from Liu (2008) and this paper. Left, relative differences in mass. Right, relative differences in extinction (solid lines), absorption (dashed lines) and back-scattering ($\sigma_{\mathrm{b}}$; dot-dashed) cross-sections. Shown as functions of $D_{\mathrm{max}}$ and using the data from Liu (2008) as reference (i.e. mass differences calculated as $(m - m_{\mathrm{Liu}})/m_{\mathrm{Liu}}$ etc.). The comparison was made for one temperature (233.15 K) and sizes covered by Liu (2008).

the decision to place dipole centres at $\ldots, -l, 0, l, \ldots$ or $\ldots, \frac{-l}{2}, \frac{l}{2}, \ldots$ (where $l$ is the dipole size) has here a significant impact, both for the effective mass after discretisation and the final optical properties.

The differences in the mass of sector snowflakes give a direct imprint in the extinction reported for this habit by the two databases (Fig. 8), but it is also noticeable that the difference in extinction is for some sizes considerably larger than the mass

difference. However, the most striking feature is the high deviation in back-scattering. Again, the sector snowflake stands out, and, as a much more representative example, a comparison for the block column habit is shown in Fig. 9. The agreement is better for the block columns in overall terms, but the back-scattering still frequently deviates with 10% or more.

A comparison with data of Hong et al. (2009) is found in Fig. 10. Clear systematic differences originating from the different refractive index models applied can be recognized. The scattering data of Hong et al. (2009) are based on the refractive index

of Warren (1984). This parameterisation deviates significantly from more recent ones (Sec. 4.1), particularly regarding the imaginary part. The deviations in refractive index are such that the data of Hong et al. (2009) exhibit a higher/lower absorption below/above about 400 GHz. The difference in the real part of the refractive index is much less pronounced and there is a fairly good agreement on the extinction due to scattering. The same is true for the back-scattering at 90 GHz, while the back-scattering at e.g. 664 GHz deviates significantly for mm-sized particles. Putting aside deviations that should be caused by the

older refractive index data used by Hong et al. (2009), the remaining data indicate a good underlying agreement between the two sets of DDA calculations.

The difference in refractive index with respect to Hong et al. (2009) is relatively small at 448 GHz. Having this in mind, the combined interpretation of Figs. 7, 9 and 10 should be that the calculation of extinction disagrees with 10% at most, but is generally inside 5%. Larger differences are found for back-scattering, particularly at higher frequencies. If only the radar




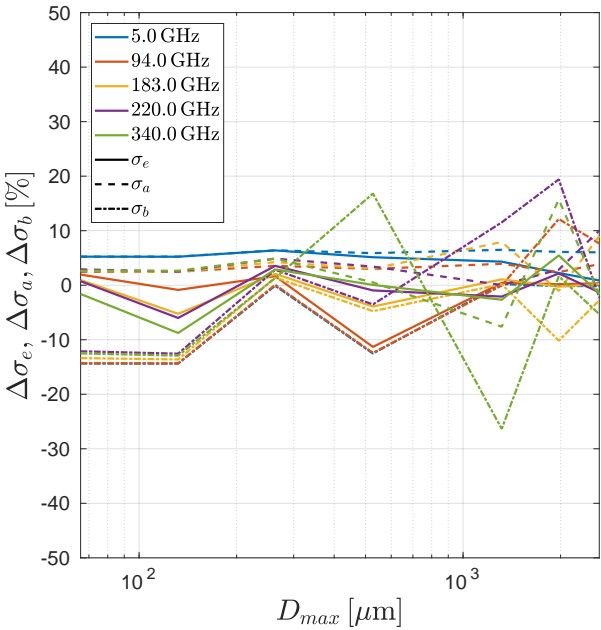

**Figure 9.** Comparison with block column data of Liu (2008). Otherwise as right panel of Fig. 8.

frequencies of today (i.e. $< 100\,\mathrm{GHz}$) are considered, then also the general agreement for back-scattering is on the order of 10%, with a strong exception in the sector snowflake habit and some data points with early stop of the Romberg scheme.

The results found for back-scattering give a direct indication on the accuracy of the phase matrix for the direct backward direction. This is likely the direction with the highest calculation errors. However, the finer details of the phase matrix should not be critical for passive measurements as the radiation field is relatively smooth (except around the limb direction), and the overall accuracy of the phase matrix should be similar to the one of the extinction. To study the effective accuracy of the phase matrix data provided would require full radiative transfer simulations, which are beyond the scope of this article.

## 5.2 Effective density

Figure 11 displays effective densities for a selection of habits. The effective density is defined as

$$\rho_{\mathrm{e}} = \frac{6m}{\pi D_{\mathrm{max}}^3}, \tag{16}$$

i.e. the mass divided by the volume of the minimum circumsphere. It is linked to the $a$ and $b$ coefficients of a habit as:

$$\rho_{\mathrm{e}} = \frac{6a D_{\mathrm{max}}^{b-3}}{\pi}, \tag{17}$$





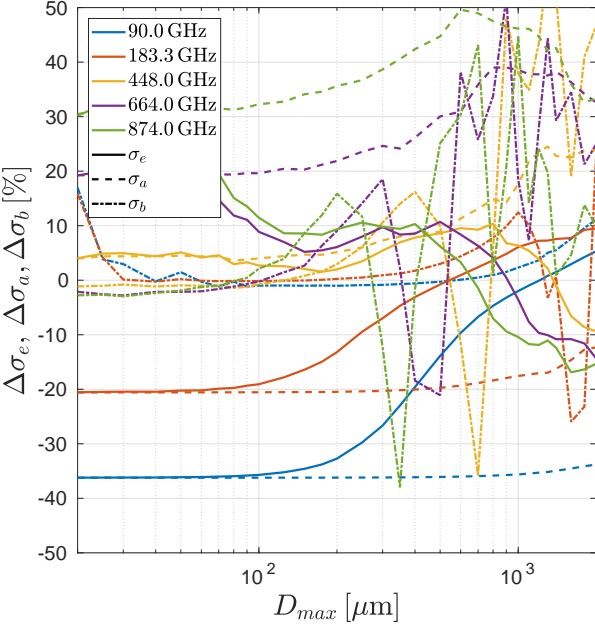

**Figure 10.** Comparison to Hong et al. (2009) of 8-column aggregate data: relative differences in extinction (solid lines), absorption (dashed lines) and back-scattering (dot-dashed). Differences reported as $(\sigma - \sigma_{\mathrm{Hong}})/\sigma_{\mathrm{Hong}}$. The comparison was made for one temperature (243.15 K) and sizes covered by Hong et al. (2009).

where the $m$ has been replaced by Eq. (5). Accordingly, the data in Fig. 11 should ideally end up along straight lines. This is also the case for habits having an analytic description of the shape, while deviations are observed for habits having stochastically generated shape data.

The effective density is governed by two main factors, the particle's "fluffiness" and aspect ratio. Habits having throughout a

5    high $\rho_e$ are generally dense particles with a spherical appearance. Particles with low values, have either high aspect ratio (such as the sector snowflake) or are very porous. Habits with $b < 3$ have negative slopes in Fig. 11 and tend to grow differently in the spatial dimensions e.g. the plate type 1 habit that has an increasing aspect ratio with size. However, $b < 3$ can also correspond to particles that increase in fluffiness with increasing distance to the particles' centre, as e.g. the GEM snow habit.

The legend of Fig. 11 is divided into groups of habits, with single crystals having solid lines, aggregate and snow particles

10    dashed lines, and dense graupel or hail-type particles dot-dashed lines. The ice sphere habit serves as an upper bound, while the long column, the flat 3-bullet rosette and the sector snowflake exhibit the lowest density, depending on $D_{\mathrm{max}}$-range. More generally, the graupel and hail habits occupy the upper part of the plot, being the most dense habit group. The aggregates and crystals tend to share the same area in the middle to lower part of the graph. It will be seen later on, that the effective density has noticeable impact on the observed optical properties.



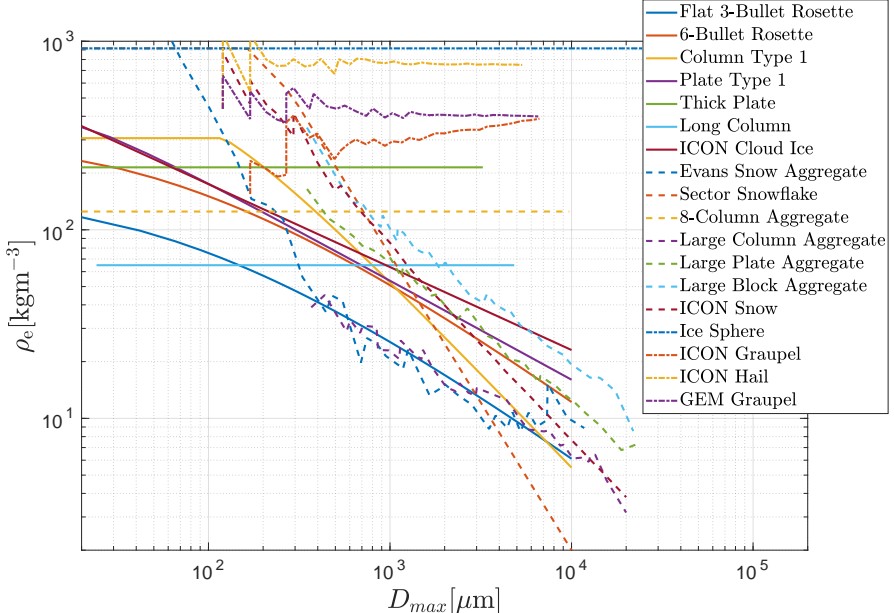

**Figure 11.** Effective densities $\rho_{\mathrm{e}}$ of selection of habits, as a function $D_{\mathrm{max}}$.

### 5.3 Extinction and back-scattering

Example data of extinction and back-scattering efficiencies are displayed in Fig. 12 for some selected frequencies and habits. The habits and legends are the same as in Fig. 11, i.e single crystals have solid lines, aggregate or snow particles dashed, and dense graupel or hail-type particles dot-dashed lines. The extinction and back-scattering efficiencies are calculated as

$$Q = \frac{4\sigma}{\pi D_{\mathrm{veq}}^2}, \tag{18}$$

where $\sigma$ is the given cross-section. That is, both $x$ and $Q$ are defined with respect to $D_{\mathrm{veq}}$. A related remark is that efficiencies are thus compared in Fig. 12 between particles having the same mass (as a given $x$ implies a specific mass).

A few characteristics can be observed in Fig. 12. Firstly, the groups tend to cluster. For example, the aggregate habits tend to have low extinction at 175 GHz and high extinction at 670 GHz. An opposite tendency can be observed for the dense particles given dot-dashed lines, that have relatively high and low extinction at 175 and 670 GHz, respectively. The crystal habits (full lines) have efficiencies similar to heavy/dense particles at 175 GHz, but varying, intermediate, extinction values at 670 GHz. Similar observations can be made for back-scattering; dense particles tend to have high back-scattering, and aggregates low back-scattering. Interestingly, the features observed for effective density in Fig. 11 are to a large degree transferred to the scattering properties in Fig 12, but exceptions are also noted. For example, the spheres and ICON hail have comparatively low extinction at $x > 3$ at 175 GHz, and several crystal habits have higher back-scattering at 88.8 GHz than the dense particles. Overall, the effective density can be considered an important parameter, but other aspects, such as exact particle shape, can




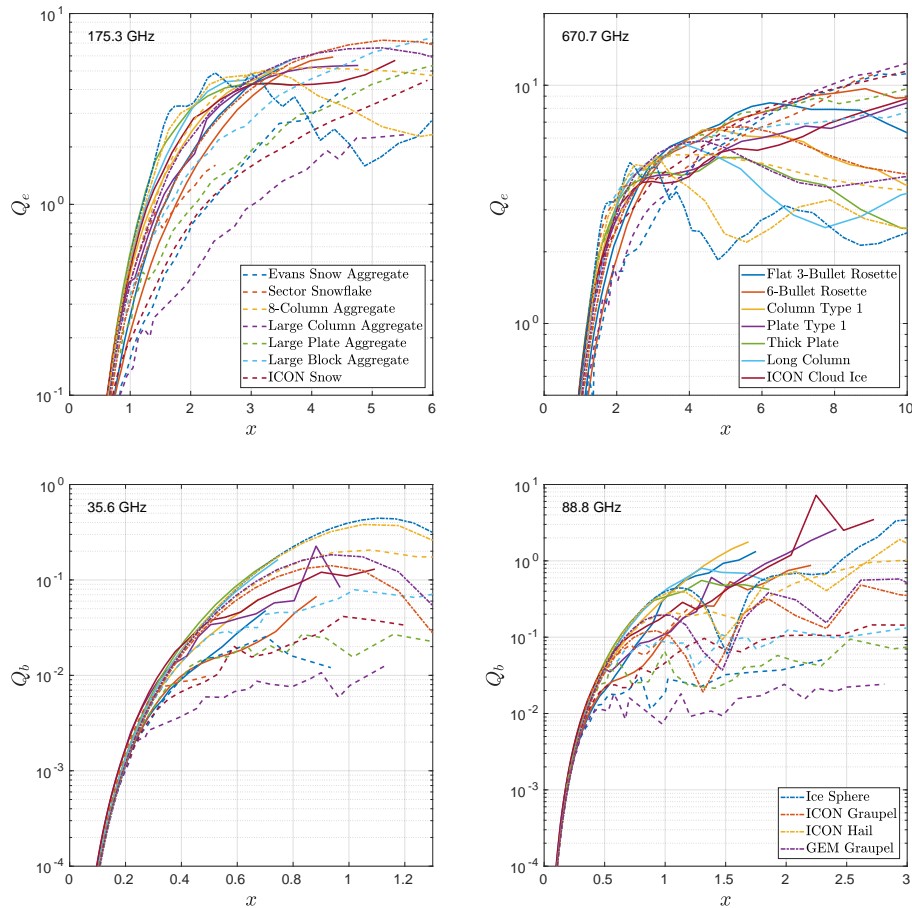

**Figure 12.** Example extinction (top row) and back-scattering (bottom row) efficiencies (Eq. 18) as a function of the size parameter (Eq. 4). Legends are valid across all panels.

influence as well. This discussion refers to how extinction and back-scattering are related to particle mass, other relationships can be found if the comparison is made with respect to e.g. $D_{\max}$.

### 5.4 Triple frequency signatures

A common manner to evaluate scattering properties in the radar community is the triple frequency signature (Kneifel et al., 2011). This is a multi-frequency bulk back-scattering analysis, involving three frequencies with data reported as two dual-wavelength ratios (DWR). The approach involves the effective reflectivity factor $Z_e$:

$$Z_e = 10^{18} \frac{\lambda^4}{\pi^5 \cdot |K|^2} \int_0^\infty \sigma_{\mathrm{b}} D_{\max} \cdot N(D_{\max}) \cdot \mathrm{d}D_{\max}, \tag{19}$$





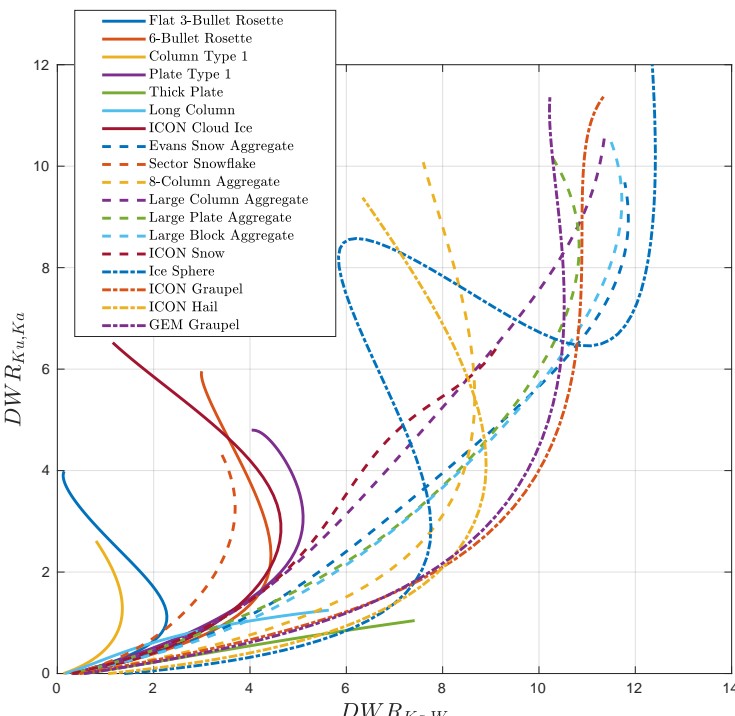

**Figure 13.** Triple frequency signatures of selected habits, assuming an exponential particle size distribution (Eq. 21) and radar frequencies of 13.4 ($K_u$-band), 35.6 ($K_a$-band) and 88.8 (W-band) GHz. The x-axis displays DWR between $K_a$ and W bands and the y-axis shows DWR between $K_u$ and $K_a$ bands.

where $\sigma_b$ is the back-scattering cross-section, $N(D_{\max})$ the number of particles of size $D_{\max}$, and $K$ is calculated as $K = (m^2 - 1)/(m^2 + 1)$, with $m$ being the complex refractive index of liquid water. The DWR for two wavelengths $\lambda$ is then obtained as

$$DWR_{\lambda_1,\lambda_2} = 10 \cdot \log\left(\frac{Z_e(\lambda_1)}{Z_e(\lambda_2)}\right). \tag{20}$$

5   Example triple frequency results are shown in Fig. 13, using three standard radar frequency bands. Refractive index of water ($m$) was taken from Ellison (2007), and an exponential PSD, defined over $D_{\max}$:

$$\frac{dN}{dD_{\max}} = N_0 \exp^{-\Delta D_{\max}}, \tag{21}$$

was assumed. $N_0$ is set to a constant value of 1, which is of no consequence since this constant is cancelled out in Eq. (20). In order to not let truncation issues influence strongly, $\Delta$ was varied for each habit separately, in such a way that the mean size of

10   the PSD ranges from 1.5 times the smallest available $D_{\max}$ to $1/3$ times the largest available $D_{\max}$.

Effective reflectivity is defined in such a way that if all particles scatter according to Rayleigh theory, $Z_e$ is independent of frequency. This has the consequence that most high slope parameters ($\Delta$), which correspond to low ice water contents, give rise





to zero or low DWR. The other end of each triple frequency curve corresponds to the highest $\Delta$ that could be accommodated with the size range available for the habit.

The triple frequency signatures have been shown to act as an indication of effective density, frequently linked to the degree of riming (Kneifel et al., 2015; Yin et al., 2017). Fig. 13 shows that the back-scattering of the database habits match the general expectations. First of all, the data spans the same general ranges of $DWR$ as reported by e.g. Kneifel et al. (2011); Kulie et al. (2014). Further, the single crystal habits (full lines), which are of comparatively low effective density (see Fig. 11) tend to occupy the left part of the graph. On the other hand, the dense graupel and hail habits (dot-dashed lines) are found in the rightmost part of the graph. The aggregate type habits generally occupy the space in between the crystals and denser habits.

## 6  Database: format and access

This section outlines database format and accessibility, while the detailed documentation is provided with the database itself.

### 6.1  Format

The database is structured in a folder hierarchy, following the habit classification displayed in Table 2. At the bottom of the folder structure, there is one folder for each habit. These folders contain auto-generated pdf documents and text files that summarise the scattering data as well as a "habit logo" figure (Figs. 1 and 2). The actual data are also found in these folders, with one netCDF4 file for each particle size of the habit. Inside each netCDF4 file, there is an internal folder structure to organise the data according to frequency and temperature. These folders contain the scattering data, as well as particle shape and processing information. The stored variables and their attributes are summarised in Table 6. The variable names should hopefully all be self-explanatory, but for clarity they are accompanied with a description tag.

The group `SingleScatteringData` contains not only the core scattering properties ($\mathbf{Z}$, $\mathbf{K}$ and $\mathbf{a}$), but also information on frequency, temperature and angular grids. The `ShapeData` group provides information on the particle behind the scattering properties, such as: mass, $D_{\mathrm{max}}$, $D_{\mathrm{veq}}$, aspect ratio, aerodynamic area equivalent diameter and refractive index. The `CalculationData` group contains ADDA settings and log data (DDA residual error used, data of completion, etc.). Particularly, an identifier of used shape file is given, and these files are part of the database. Thus, each scattering calculation is fully reproducible.

For orientation-averaged scattering properties, the optical properties exhibit features that can be used to reduce the overall data volume. For example, the scattering matrix has only 6 unique, non-zero, elements: $\mathbf{Z}_{11}$, $\mathbf{Z}_{12}$, $\mathbf{Z}_{22}$, $\mathbf{Z}_{33}$, $\mathbf{Z}_{34}$ and $\mathbf{Z}_{44}$. The extinction matrix has only one such element, $\mathbf{K}_{11}$. Similarly, only one element of the absorption vector, $\mathbf{a}_{11}$, is non-zero. Only these unique elements are stored, together with information on how full matrices and vectors shall be populated. For example, the scattering matrix is stored in the database as a tensor (`PhaMat_data`) with dimensions (`aa_scat`, `za_scat`, `aa_inc`, `za_inc` and `phaMatElem`), where `phaMatElem` are the six stored values. Information on how to reconstruct the full matrix





**Table 6.** Summary of core data format.

| | |
|---|---|
| Attributes: | `date, version` |
| Groups: | |

| | | |
|---|---|---|
| `SingleScatteringData:` | | |
| | Attributes: | `orient_type` |
| | Dimensions: | `aa_scat, za_scat, aa_inc, za_inc, scatMat_row,` `scatMat_col, phaMatElem, extMatElem, absVecElem` |
| | Variables: | `frequency, temperature, aa_scat, za_scat, aa_inc,` `za_inc, phaMat_index, extMat_index, absVec_index,` `phaMat_data, extMat_data, absVec_data` |
| `ShapeData:` | | |
| | Attributes: | `description, source, refrIndex_model, habit_file_id,` `habit_id, phase, refrIndex_homogenous_bool,` `density_homogenous_bool` |
| | Variables: | `diameter_max, diameter_vol_eq, aspect_ratio` `diameter_area_eq_aerodynamical, mass, dpl,` `N_dipoles, refrIndex_real, refrIndex_imag, alpha,` `beta, gamma` |
| `CalculationData:` | | |
| | Attributes: | `method, software, software_version, system, n_nodes,` `n_cores, date_completion, ADDA_eps,` `ADDA_avgParam_file, ADDA_scatParam_file` |

is provided separately, in `phaMat_index` that contains the position matrix

$$
\begin{pmatrix}
1 & 2 & 0 & 0 \\
2 & 3 & 0 & 0 \\
0 & 0 & 4 & 5 \\
0 & 0 & -5 & 6
\end{pmatrix}.
\tag{22}
$$

The numbers corresponds to the indices of `phaMatElem` in `PhaMat_data`, i.e. where to place those values in the matrix. The indexing is 1-based, zero flags that the matrix is zero at that position. A negative number means the the database value

5    shall be multiplied with -1 before it is inserted in the matrix. Corresponding systems are used for the extinction matrix and the absorption vector.



## 6.2 Interface and standard habits

The scattering data described above can easily be extracted using any netCDF or HDF5 interface. However, for enhancing the user friendliness, two data interfaces are distributed with the database. The interfaces are implemented in MATLAB and Python. The aim has been to keep the functionality as similar as possible between the two interfaces (but details of the pro-
gramming languages used make some differences unavoidable). The MATLAB interface should function with any relatively new MATLAB version. The Python interface should function with both Python 2 and Python 3, but Python 3 ($\geq 3.5$) is recommended for full functionality. There are no extensive requirements on the interfaces. The Python interface requires the `netCDF4`, `numpy` and `os` packages, and the MATLAB interface the `netcdf` package.

The functionalities provided by the two interface are: exploration the database content, extraction of selected parts of the
data (with respect to habit, size, frequencies and temperature), compiling imported data into a more compact data format, interpolation of data in temperature, frequency, size and angles, preparation of habit mixes and calculation of bulk properties.

The interfaces provide also support for converting data to the format used by the ARTS (Eriksson et al., 2011a) and RTTOV-SCATT (Bauer et al., 2006) forward models. Both interfaces can deal with the ARTS scattering format, on the condition that the Atmlab (Matlab) and Typhon (Python) packages[4] are installed. Only the Python interface supports RTTOV, i.e. conversion
of the database scattering properties to data tables that can be digested by RTTOV-SCATT's Mie coefficient module. (This requires an update of the Mie coefficient module developed along with the ARTS database. This will be part of future RTTOV releases, but is also available as patch file provided with the database.)

To allow a quick introduction to the database, a set of "standard habits" has been generated and provided in both the ARTS and RTTOV-SCATT formats. These habits should cover a range of conditions, such as cloud ice, snow, graupel, and hail. Most
of these habits are a direct compilation of the database content, but some can be denoted as habit mixes. These mixes were created to simplify the usage of some of the aggregate and rimed habits that do not give full coverage in size. In order to also cover the lower size range, those habits were complemented with data from some crystal type. Further details about the standard habits are provided in the accompanying `ReadMe` file.

## 7  Data availability

The database and its interfaces are publicly available at Zenodo, a free of charge research data repository hosted by CERN. The interfaces and the database itself are available under different uploads due to different licensing, with separate DOIs. The database DOI is https://doi.org/10.5281/zenodo.1175572 and the interfaces DOI is https://doi.org/10.5281/zenodo.1175588. This article refers to the data stored at Zenodo. As the database will be extended gradually, there is also a "development version" where the new data will be added until a new official version is released. This version is placed at a ftp-server hosted
by Hamburg University, accessible through the ARTS homepage[5]. The database itself is provided under the CC BY-SA licence

---

[4]Both available at http://www.radiativetransfer.org/tools/
[5]http://www.radiativetransfer.org/tools/



[6], allowing the user to share and adapt the material, under the conditions that appropriate credit is given and and indication of any changes made is given. Also, distribution of any modified or built-upon content must be done under the same licence. All source code falls under the GPL[7] (General Public License).

## 8 Conclusions

The first version of a general, and fully publicly available, database of hydrometeor optical properties is presented. The database covers frequencies of 1 to 886 GHz, i.e. the full microwave region is targeted. This initial version is restricted to totally random particle orientation and focuses on particles consisting solely of ice. A variety of particle shapes, organised into "habits", are provided, in order to allow representation of cloud ice, snow, graupel and hail in radiative transfer simulations. In total, 34 ice hydrometeor habits are provided, and, as complement, data for liquid spheres are included. Considered temperatures are 190, 230 and 270 K for ice, and 230, 250, 270, 290, and 310 K for liquid water. The data fully support polametric applications of both passive and active type.

The "ARTS database" presented here is the most extensive one at hand for totally random orientation. All beside one of the existing databases are also restricted to this particle orientation case. Only some main remarks are given here, for a review see Sec. 1 and Ekelund et al. (2017). Only Ding et al. (2017) offer the same wide frequency coverage. In comparison to that data, the ARTS database includes a considerably higher number of habits (34 vs. 12) and a somewhat higher coverage in particle size. As all the aggregate habits in Ding et al. (2017), as well as in Hong et al. (2009), have $b = 3$ (see Eq. 5), it can be claimed that the ARTS database provides the first publicly available realistic representation of snow aggregates at microwave frequencies above 200 GHz. For lower frequencies, a broader coverage of snow is provided by the database of Kuo et al. (2016).

The aggregates of Ding et al. (2017) can potentially work as a proxy for hail and heavily rimed particles. The ARTS database contains a more diverse set of particle shapes for these classes, but covers also the 8-column aggregate part of both Hong et al. (2009) and Ding et al. (2017). A large part of the effort behind the database is the development of tools to generate realisations of these particle types. Regarding column and plate ice crystals, it was noted that such habits in Liu (2008) all have $b = 3$, and for these particle categories the data provided with the variation of aspect ratio taken from Hong et al. (2009) should be more realistic. The "update" of the data of Hong et al. (2009) using a better parameterisation of refractive index resulted in changes up to 50% (Fig. 10), as well as a broader coverage in frequency, temperature and sizes.

All optical properties, except some reference data for spheres calculated by Mie-code, were derived by the discrete dipole approximation (DDA), handling arbitrarily shaped particles. Considering the high calculation burden of DDA (particularly at higher size parameters), it was not possible to both provide extremely accurate data and a broad database. It was decided at this stage to aim for a moderate accuracy, in order to first get a relatively broad overview. The basic idea is to explore the general properties of the data generated (manuscript in preparation), in order to better understand what particle shape properties govern the optical properties. It is not feasible, or desirable, to have data for every possible particle shape, but at this moment it is not

---

[6]https://creativecommons.org/licenses/by-sa/4.0/
[7]http://www.gnu.org/licenses/gpl-3.0.en.html





clear to what extent for example micromorphological aspects like the shape and size of the crystals inside an aggregate need to be considered.

Comparisons to T-matrix, Liu (2008) and Hong et al. (2009) indicate that accuracy of the data is in general within 10% (Sec. 5.1). This potential inaccuracy must not be ignored, but should today be relatively small compared to all other uncertain-
ties involved in full simulations and retrievals. For example, Baran et al. (2017) reasoned in the same way and were also content with a 10% accuracy. However, much higher discrepancies were found occassionally, particularly regarding back-scattering. Further tests will be performed to better characterise the accuracy of the optical properties provided by the database, and, if found necessary, e.g. for typical radar frequencies, data will be recalculated with more demanding settings to provide more accurate back-scattering.

The calculation burden of DDA resulted also in that a cut-off around size parameter ($x$) 10 was introduced, but it is argued that this in general has an insignificant or low impact on bulk properties (Sec. 2.5). In other works, the need for calculations to much higher $x$ is discussed, but it is here essential to notice how $x$ is defined. We calculate $x$ based on volume equivalent diameter ($D_{\mathrm{veq}}$), while e.g. in Baran et al. (2017) it is defined with respect to maximum diameter ($D_{\mathrm{max}}$) resulting in a higher $x$ for a given particle (except for spheres).

Besides accuracy improvements, the next version of the database is planned to also incorporate data for melting and moderately rimed particles. A more major addition will be to include data for oriented particles. Such data are today very sparse and limited to a few frequencies below 200 GHz (e.g. Tyynelä and Chandrasekar, 2014). Initial calculations for "azimuthally random orientation" have been performed, with the conclusion that these calculations are much more demanding than the totally random case, with respect to both calculation burden and final data storage. Thus, practical hurdles must be overcome, and it
is essential to carefully analyse the data at hand to guide the future calculations towards the most representative habits, as well as determining how many frequencies, temperatures and sizes the database must cover to be of practical use.

*Author contributions.*  PE initiated the database, has acted as project leader, coded RimeCraft and the Matlab interface, and written article text. RE has produced all data, designed the database structure and written article text. JM developed the RTTOV and Python interfaces, provided feedback on data and methods, and contributed parts of the article text. MB, OL and SAB participated in planning of the database,
have contributed to data and text, and started the production of the data for oriented particles, that will be a main addition of the next database version.

*Competing interests.*  The authors declare that they have no conflict of interest.

*Acknowledgements.*  A large part of the database and associated tools were produced inside a study funded by EUMETSAT (Contract No. EUM/COS/LET/16/879389). The study manager at EUMETSAT was Christophe Accadia, that provided appreciated feedback and inspira-
tion. Remaining funding for P.E., R.E. and J.M. was provided by the Swedish National Space Board. S.A.B. was partially supported through the Cluster of Excellence CliSAP (EXC177), Universität Hamburg, funded by the German Science Foundation (DFG), through BMBF

project HD(CP)$^2$, Fkz. 01LK1502B and Fkz. 01LK1505D, and through the HALO/HAMP DFG project (BU 2253/3-1). Special thanks go to Torbjörn Rathsman, who implemented the SnowFlake toolkit as his master thesis project. We also thank Frank Evans and Jani Tyynelä for kindly providing shape data. The RTTOV interface was designed with input from Alan Geer at ECMWF. ADDA calculations have been done at central computing facilities at both involved universities (C3SE/Chalmers and DKRZ(PID 878)/Hamburg). CERN is thanked for

5   providing data storage.



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
