# Peer review of "A general database of hydrometeor single scattering properties at microwave and sub-millimetre wavelengths"

_Earth System Science Data, 2018_

## Referee Comment (RC1) · Anonymous Referee #1 · 23 May 2018

General comments:

This manuscript introduces a database of scattering properties for different ice particles including also raindrops. This database is one of the first ones to include high microwave frequencies up to the submillimeter range. The authors' goal is to provide an easy interface for users and a broad range of particle types, which is important in many retrieval studies. In particular, these can be easily utilized in the atmospheric radiative transfer simulator (ARTS).

Overall, I find this manuscript and the database highly relevant for researchers wanting to include better microphysical models in their algorithms.

[Figure]

I recommend the publication of this manuscript with only minor revisions.

Specific comments:

page 14, line 16, sentence 'Contrary to most aggregate...': I find this a bit too general statement. The more physics-based stochastic aggregation model which takes into account the mutual velocities and random collisions of falling ice crystals and aggregates, such as the one described by Westbrook (2004) and Maruyama (2005), is used in many models, e.g. Kuo et al. (2016), Leinonen and Szyrmer (2015) and Tyynelä and Chandrasekar (2014).

page 14, line 19, sentence 'Aggregate sticking is only...': The assumption of parallel faces is generally not assumed in the physics-based aggregation models due to the turbulence close to the particles (Pruppacher and Klett, 1997). This is also assumed in Maruyama's aggregation model. I would like to see what this decision is based on. Also typo: 'be be parallel'.

page 17, line 2: This sentence needs some revision. There are no interactions between particles in DDA. You are computing the interaction between dipoles inside a single particle. Amplitude matrix and Mueller matrix elements have in principle the same information included in them in a fixed orientation. When you do the orientation averaging scheme as you described in Eq. (2), you are summing squared electric fields from different orientations/particles together, which means that the phase information of the scattered electromagnetic waves for individual orientations is lost. As an example, some radar measurenents, like the specific differential phase Kdp, can capture the phase information. In this case, you need to sum the electric fields directly in the forward scattering direction for each orientation and particle.

page 21, line 16, sentence 'The Tyynelä aggregates...': Here it might be simpler to just mention that the aggregation was based on Westbrook (2004), which is a similar model as Maruyama et al. (2005).

[Figure]

page 30, Fig. 13 caption: Why 88.8 GHz? The W-band cloud radar frequency is 94 GHz, not 88.8 GHz. Radiometers use the 89 GHz channel.

References:

Pruppacher, H. R., and J. D. Klett (1997), Microphysics of clouds and precipitation, Kluwer Academic, 954 pp.

Westbrook, C.D. (2004), Universality of snow formation. PhD Thesis, University of Warwick, Coventry, UK.

---

## Referee Comment (RC2) · Anonymous Referee #2 · 29 May 2018

This work describes an important contribution to our capability in using electromagnetic scattering properties of ice particles to learn more about them in Earth's atmosphere. It is certainly worthy of publication in ESSD. It is recommended for publication after the authors deal with the suggested minor revisions below.

After writing out all of the suggested minor revisions below, perhaps the biggest gift the authors can give readers of their article is described in the sentence at the end of comment 20 below:

"Even though the paper is long, the authors need to go over every sentence within it time and again so as to get them all in as good as shape as is possible."

[Figure]

Suggested Minor Revisions:

0) I wrote comments on the manuscript as I read through it. As some of them may be of value to the authors, I am returning the marked-up manuscript to the authors for their consideration.

1) The authors use of "e.g." throughout is interesting. Consider Line 8 of the abstract where the phrase

"and remaining habits are aggregates of different types, representing, e.g., snow and hail"

occurs. In this instance "e.g." is used not only to demarcate a list but also to assist in the transition started with "representing." In this case is the word "representing" really necessary? I think the word "representing" can be left out:

"and remaining habits are aggregates of different types, e.g., snow and hail"

This happens in so many places throughout the text that I thought it worthwhile for the authors to think about it.

2) Page 1, Lines 18-20: The part of the sentence after the "and" has nothing to do with the part of the sentence before the "and". As a result, this sentence would appear to me as a run on sentence whose logic is not completely clear. This happens a lot throughout the manuscript and the manuscript would be a lot more pleasant to read if these types of sentences were eliminated.

3) Page 2, Lines 11-12: "Extinction due to scattering is the main process by which rain and ice particles are sensed" Is this a generally true statement? What about emission from rain at frequencies for which the underlying surface has a low emissivity?

4) Page 4, Lines 14-15: "but already this version is more extensive than earlier datasets"

As the database currently stands, it will not be used that much for radar applications

because it contains results only for randomly oriented particles and the value of polarimetry is limited for such particle orientations. For example, ZDR is always 1 (or 0 in units of dB) for randomly oriented particles but in nature it is often much different from 1 (or 0 dB); as a result, the current database has nothing to offer in terms of ZDR. Only when the database comes to include results for oriented particles might the statement above become true.

5) Page 6, Line 21: What is a "subscribing sphere"? Do you mean "circumscribing sphere" here?

6) Page 6, Lines 22-23: Dmax is not ambiguous at all if it is defined as the maximum distance between any two points within an ice particle.

7) Page 6, Line 24: "the mass or Dveq should in general be preferred"

Knowing the mass of a particle is always important. But why the "should" in "Dveq should in general be preferred"? Perhaps Dveq is preferred in your application but it is not so clear why it should be preferred in general.

8) Table 3 brings no value to what is already contained in Table 1. Removing Table 3 does no harm to the manuscript.

9) Page 7, Line 21: "The top category is 'phase'" and yet in Table 2 the top category is "Orientation"; the text is inconsistent with the table and "Orientation" within Table 2 is never considered in the text. Can this row of Table 2 be removed?

10) Figure 1j: The "Gem ice cloud" looks like a sphere but on Page 20, Line 15, it is described as being a spheroid. Can you make it look a bit like a spheroid in Figure 1j for the sake of consistency and to distinghuish it from Figure 1h?

11) Page 12, Line 14: Beginning to notice lots of occurrences of "should". Better in scientific writing to stay away from "should" as this word does not mean much in such writing.

12) Page 14, Lines 19-26: The description about aggregation is confusing. How can "modelled as if the two particles collide with each other at random angles" be consistent with "the two involved surface normals are forced to be parallel"? If "face to face sticking is ensured" the particles would be flat but the figures do not show flat aggregates. Something is awry here.

13) Page 15, Line 14: "to take the temperature gradient into account"

This phrase has no context. Where did it come from and why is it important?

14) Page 16, Line 29: The Mueller matrix is Vector M and the 11 element of of the Mueller matrix is Vector M sub 11. But on Page 18, Line 1, the 11 element of vector S is not Vector S sub 11; it is simply S sub 11. This is inconsistent. Using bold fonts for matrix elements seems unusual. From this perspective Page 18, Line 1, might be the more conventional way of expressing elements. Nonetheless, make all consistent here.

15) Page 17, Line 7: What does "More negative sigma sub a" mean? Not clear.

16) Figure 6: The legend on this figure is never defined nor is the symbol dBT.

17) Page 22, Lines 2-3: "with their faces against each other"

This is related to comment 12) above. This would seem to indicate flat aggregates but the aggregates are not flat so same misunderstanding as before.

18) Page 22, Line 4: "the whole set of particles"

What comprises a "whole set of particles"? This is not quite clear from the text just before it. On Page 21, Line 30, the statement "six simulations in total were performed" which would seem to imply six distinct particles being created. Perhaps this should be "six sets of simulations in total were performed" in which each set is for a habit and the elements of the set for a habit are particles of different sizes? Is this correct? What is going on here is not quite so clear.

19) Page 22, Line 20: What does "15%" mean on this line? 15% of some quantity such as the density of solid ice? And if so, why so low as compared to the densities of graupel in Figure 11? The meaning of this "15%" is confusing.

20) Page 23, Lines 14-15: "DDA can shift the pattern somewhat in size, causing the data for a specific size to deviate signifantly"

It is this type of sentence that hurts the quality of the manuscript. What does "DDA can shift the pattern somewhat in size" mean? The word "somewhat" is vague and how does DDA shift anything? DDA is an algorithm that computes values and does not move anything around. This phrase also contains a misspelled word that would be caught by a spell checker. Even though the paper is long, the authors need to go over every sentence within it time and again so as to get them all in as good as shape as is possible.

21) Page 28, Lines 8-16: The line plots within the four subpanels of Figure 12 are being compared on these lines of text, yet not a single x-axis or y-axis are identical amongst the four subpanels. This makes comparing the lines difficult. Is there a way to treat the x- and y-axes of these four subpanels to make comparisons between them easier?

22) Page 29, Eq. 19: This equation should contain sigma_B(D_max); that is, the D_max must be in parentheses.

23) Page 30, Lines 5-6: Why even bring up the topic of liquid water refractive indices when liquid water results do not occur anywhere within Figure 13?

24) Page 31, Line 30: Should the second "phaMatElem" be "phaMat_data"? Note that "PhaMat_data" occurs in the text while "phaMat_data" occurs in the table.

Please also note the supplement to this comment:
https://www.earth-syst-sci-data-discuss.net/essd-2018-23/essd-2018-23-RC2-supplement.pdf

[Figure]

[Figure]

**Supplement:**

[revised manuscript text omitted]

---

## Referee Comment (RC3) · I. Adams (Referee) · 5 Jun 2018

**General Impressions**[1]

The manuscript "A general database of hydrometeor single scattering properties at microwave and sub-millimetre wavelength" offers specifics on the most detailed and wide-ranging scattering database currently available. This dataset is a valuable resource to the radiative transfer community. The decisions and resulting limitations are
* * *
[1]Given that I am a user of this dataset who has evaluted the current version and previous beta versions, and a user of and contributor to the ARTS model, for the sake of transparancy, I am foregoing anonimity for this review.

[Figure]

explicitly stated and justified, e.g., limiting size parameter for computational reasons or the choice of liquid water permittivity model, and the uncertainty analysis is forthcoming. My primary issues deal with some of the analysis concerning radar applications, and the paper requires a bit of copy editing. Otherwise, this manuscript should be published once the detailed comments below are addressed.

**Specfic Comments**

Page 2, lines 11-12: The suggestion here is that the optical properties necessary for passive radiative transfer do not apply to active. There are numerous cases where more than just reflectivity is necessary. First , for all but the longest wavelengths in the microwave regime, extinction will impact observed reflectivities. In practice, radars transmit polarized radiation, where $I = 1$ and any other single element is either $+1$ or $-1$. For example, precipitation radar are often polarized horizontally $Q = -1$ and/or vertically $Q = 1$ with respect to some reference frame. This results in the entire top-left block of the phase matrix (terms 11, 12, 21 and 22) being required for reflectivity, differential reflectivity ($Z_{hh}/Z_{vv}$), and linear depolarization ratio (e.g., $Z_{vh}/Z_{vv}$). Higher order terms of the Stokes vector, phase matrix, and extinction matrix are necessary when considering reference frame rotations (3D RTM), other polarimetric radar variables ($\rho_{hv}$, $K_{dp}$), or multiple scattering effects. Multiple scattering can be observed at a wide range of frequencies, including at X-band in the presence of hail (Battaglia et al., 2016).

Page 5, line 31: Azimuthally-random could also apply to cases where $\beta$ is not a Dirac delta function, e.g., for an arbitrary flutter distribution with a mean canting angle of zero.

Page 6, line 14: Specify the value used for $\rho$.

Page 7, line 21 and Page 9, Table 2: Why is "melting" included when it is not in the database? Azimuthally-random is mentioned in the text, but has been excluded from

this table. Melting should be removed.

Page 8, Table 1: Why do the liquid spheres go to such a large size? This is unphysical, since drops break up by about 8 mm in diameter.

Page 10, line 10, through page 12, line 9: The discussion of size parameter limit lacks any consideration of radar reflectivity. In the Rayleigh regime, reflectivity is proportional to the sixth moment of the size distribution, and while this $D^6$ dependence diminishes at larger size parameters, there are still significant contributions from larger particle sizes that can offset the lower probabilities. Plots similar to the extinction plots (Fig. 3 and Fig. 4) would be useful for understanding the size parameter limit with respect to reflectivity.

Page 12, line 11: In reference to Table 4, I count 35 frequencies.

Page 17: When calculating phase or matrix, a good consistency check is comparing $Z_{12}$ and $Z_{21}$, with the caveat that these terms can be signficantly smaller, numerically, than $Z_{11}$. Since these two terms should be equal for randomly oriented particles, a comparison can provide a check of the orientational averaging.

Page 22, line 19: Standard deviation is not typically used to describe a gamma distribution.

Pages 29 through 31, Section 5.4: The discussion of triple frequency signatures should use $D_{veq}$ or $D_e$ the equivalent liquid (or mass) diameter for the size distribution, instead of $D_{max}$. An exponential distribution in $D_{max}$ space becomes a much more complicated modified gamma distribution in $D_e$ space (Petty and Huang, 2011). By using $D_e$, the slope parameter $\Delta$ in (19) is guaranteed to be consistent for all particle habits, with mass being equal when the size distribution is integrated over the same size limits.

Page 30, lines 5-6: What is the reference temperature used?

**Technical Items (typos, grammar, spelling, etc.)**

The list below should not be considered fully inclusive, and the authors should review the manuscript carefully.

Page 1, line 9: "diameter" → "diameters"

Page 2, line 30: The entire sentence is a bit awkward.

Page 3, line 2: "on the same time as" → "and"

Page 3, line 4: Awkward. Suggestion: Move "that" to between "simulations" and "can"

Page 3, line 8: Awkward. Suggestion: Remove "that" and change "so far has been" to "being"

Page 3, line 12: Remove the comma from after "role"

Page 3, line 13: Change "that" to "those"

Page 4, line 1: "There is" → "There are"

Page 5, line 31: "tilt angles" should be singular

Page 6, line 2: "for next" → "for the next"

Page 14, Subsection Heading: Capitalize "toolkit"

Page 14, line 19: "off-sets" → "offsets"

Page 14, line 27: Move "also" to between "is" and "handled"

Page 14, line 28: "should be to generate" → "is for generating"

Page 14, line 29: "on to" → "onto"

Page 15, line 12: Remove comma from between "gridded" and "particle"

Page 21, line 6: Move "third party" to before "Aggregate"

[Figure]

Page 21, line 8: "later" → "latter"

Page 21, line 13: "kept track off" → "tracked"

Page 21, line 14: "generation of snow aggregate" → "aggregation" or "simulated aggregation"

Page 21, line 22: "tool-kit" → "Toolkit"

Page 21, line 23: "hexagonals" → "hexagons"

Page 22, line 16: "toolkit" → "Toolkit"

Page 27, line 13: Remove the comma from between "on" and "that"

Page 28, line 4: "back-scattering" → "backscattering"

Page 29, line 5: "back-scattering" → "backscattering"

Page 35, line 9: "back-scattering" → "backscattering"

Page 35, line 10: "cut-off" → "cutoff"

**References**

Battaglia, A., Mroz, K., Lang, T., Tridon, F., Tanelli, S., Tian, L., and Heymsfield, G.M. 2016: Using a multiwavelength suite of microwave instruments to investigate the microphysical structure of deep convective cores. *JGR: Atmos.*, **121**, 10.1002/2016JD025269.

Petty, G., and Huang, W. 2011: The modified gamma size distribution and nonspherical particles: key relationships and conversions. *J. Atmos. Sci.*, **68**, 10.1175/2011JAS3645.1.

---

## Author Comment (AC1) · 25 Jun 2018

First of all, we thank the two anonymous referees and Ian Adams for taking time to review our manuscript and for the constructive feedback provided. The comments include fair criticism. Besides the various points raised, we are happy to notice that all three referees find a value in the database produced and that a general recommendation for publication is given.

In summary, we see a reason in basically all comments and they will be carefully considered in our revision. There are some main themes in the criticism and at this point we mainly comment on a general level.

[Figure]

The main part of the criticism deals with grammar problems and unclear language. We will do our best to improve on these aspects.

Some of the presentation issues are associated with our plans to make use of ESSD's "living data" process. Our understanding is that the ESSD article can be updated when we present new versions of the database, such as when including data for oriented particles. We plan to make use of this nice feature, and not writing a completely new article for database version 2 etc. For this reason some discussion is a bit broader than motivated by this database version, and features not yet used are mentioned (such as the database is planned to have a "melted" category). By making the presentation a bit broader, we wanted to both indicate that we have clear plans for extensions of the database and avoiding to end up with a text that has to be totally rewritten for the next database version. We understand that the text shall correctly reflect the current database version and will adjust the text accordingly, but we wanted to explain the reasoning behind our presentation approach.

This issue discussed in the paragraph above is most apparent with respect to active measurements. It's clear that the present data have restrictions with respect to radar applications, but as we have a special interest in synergy between active and passive microwave observations we wanted to include discussion of radar applications from start. Again, we will revise the text to remove unclarities.

Some response on a more detailed level:

* Ian Adams points out that 50 mm rain drops do not exist and found our choice to include such particles to be unphysical. It's true that rain drops break up when reaching a size of about 8 mm. On the other hand, drop size distributions applied do normally not consider this physical limit. In fact, they predict the presence of drops up to infinite size. For this reason, we included unrealistically large drops to allow the database user to integrate properties up to very high drop sizes (to evaluate the contribution from the unphysical size range). We would have preferred to offer a very broad size

coverage for all database habits (also suggested by Alan Geer inside the EUMETSAT study supporting the database development), but the use of DDA limited what we could achieve.

* The calculation of effective radar reflectivity (Eq 19) is defined in such way that the K-factor shall be set following the refractive index of liquid water, even if it is known that the backscattering is caused by ice hydrometeors. (Response to Referee #2)

* Referee #1 makes the comment that face-to-face sticking is generally not assumed in aggregation models and asks for an explanation. First of all, this is partly a matter of allowing overlap or not in the aggregation of the crystals. We make use of compact hexagonal crystals in our simulations, which are not easily deformed. Hence, it is questionable if significant crystal overlap in real aggregates is realistic. While dendrites are known to aggregate with the help of mechanical interlocking, faceted crystals tend to stick at surfaces, depending on electrostatic forces, surface melting or roughness (Hobbs et al., 1974). Admittedly, the no-overlap condition also makes the calculations of the aggregate volume more straightforward. Bear in mind that the particles are represented by polygon meshes in the simulations, and calculating the volume of overlapping crystals is therefore not trivial and would slow down the computations unless serious approximations are made. Hence, the decision is based on the belief that this assumption is valid for compact and pristine crystals, and the fact that it makes the aggregation simulation less complex. For aggregation of dendrites, this constraint would indeed make less sense.

References

Hobbs, P. V., S. Chang, and J. D. Locatelli (1974). "The dimensions and aggregation of ice crystals in natural clouds". J. Geophys. Res. 79.15, pp. 2199–2206.